# Role of PDGFRA⁺ cells and a CD55⁺ PDGFRA^Lo fraction in the gastric mesenchymal niche

Elisa Manieri [1,2], Guodong Tie[1], Ermanno Malagola [3], Davide Seruggia [4,5,6], Shariq Madha[1], Adrianna Maglieri [1], Kun Huang [7], Yuko Fujiwara[4,8], Kevin Zhang[4], Stuart H. Orkin [4,8,9], Timothy C. Wang[3], Ruiyang He[1], Neil McCarthy [1,2] & Ramesh A. Shivdasani [1,2,9] ✉

PDGFRA-expressing mesenchyme supports intestinal stem cells. Stomach epithelia have related niche dependencies, but their enabling mesenchymal cell populations are unknown, in part because previous studies pooled the gastric antrum and corpus. Our high-resolution imaging, transcriptional profiling, and organoid assays identify regional subpopulations and supportive capacities of purified mouse corpus and antral PDGFRA⁺ cells. Sub-epithelial PDGFRA^Hi myofibroblasts are principal sources of BMP ligands and two molecularly distinct pools distribute asymmetrically along antral glands but together fail to support epithelial growth in vitro. In contrast, PDGFRA^Lo CD55⁺ cells strategically positioned beneath gastric glands promote epithelial expansion in the absence of other cells or factors. This population encompasses a small fraction expressing the BMP antagonist *Grem1*. Although *Grem1*⁺ cell ablation in vivo impairs intestinal stem cells, gastric stem cells are spared, implying that CD55⁺ cell activity in epithelial self-renewal derives from other subpopulations. Our findings shed light on spatial, molecular, and functional organization of gastric mesenchyme and the spectrum of signaling sources for epithelial support.

Long-lived gastric and intestinal stem cells produce distinct differentiated cell types within stereotypic monoclonal crypts (intestine) or glands (stomach). *Lgr5*-expressing intestinal stem cells (ISCs) at the base of each crypt drive brisk cell turnover[1,2]. The epithelium of the gastric antrum, the region nearest the intestine, lacks villi but turns over almost as rapidly as intestinal epithelium; each gland consists mainly of mucus-producing foveolar cells that line luminal "pits" and distinct mucous cells at the base. Like ISCs, antral epithelial stem cells lie near the gland base (Fig. 1a), express *Lgr5*, and replicate symmetrically[3–6]; it is, however, unclear whether *Lgr5* marks dedicated multipotent cells or basal mucous cells with facultative stem-cell properties[7]; other candidate stem cell genes, such as *Vil1*, *Runx1*, *Sox2*, *Axin2* and *Aqp5*, express in partially overlapping compartments[4,7–10] and may also mark facultative or fractions of active stem cells. Diverse observations nevertheless support the view that multipotent antral stem cells reside near gland bottoms and that their progeny mainly migrate upward to replenish the physiologic attrition of terminal foveolar cells (Fig. 1a)[3,4].

[1]Department of Medical Oncology and Center for Functional Cancer Epigenetics, Dana-Farber Cancer Institute, Boston, MA 02215, USA. [2]Department of Medicine, Harvard Medical School, Boston, MA 02115, USA. [3]Division of Digestive and Liver Diseases, Department of Medicine and Irving Cancer Research Center, Columbia University Medical Center, New York, NY 10032, USA. [4]Department of Hematology, Boston Children's Hospital, Boston, MA 02115, USA. [5]St. Anna Children's Cancer Research Institute, Vienna, Austria. [6]CeMM Research Center for Molecular Medicine of the Austrian Academy of Sciences, Vienna, Austria. [7]Molecular Imaging Core and Department of Cancer Immunology and Virology, Dana-Farber Cancer Institute, Boston, MA 02215, USA. [8]Howard Hughes Medical Institute, Boston, MA 02115, USA. [9]Harvard Stem Cell Institute, Cambridge, MA 02138, USA. ✉e-mail: ramesh_shivdasani@dfci.harvard.edu

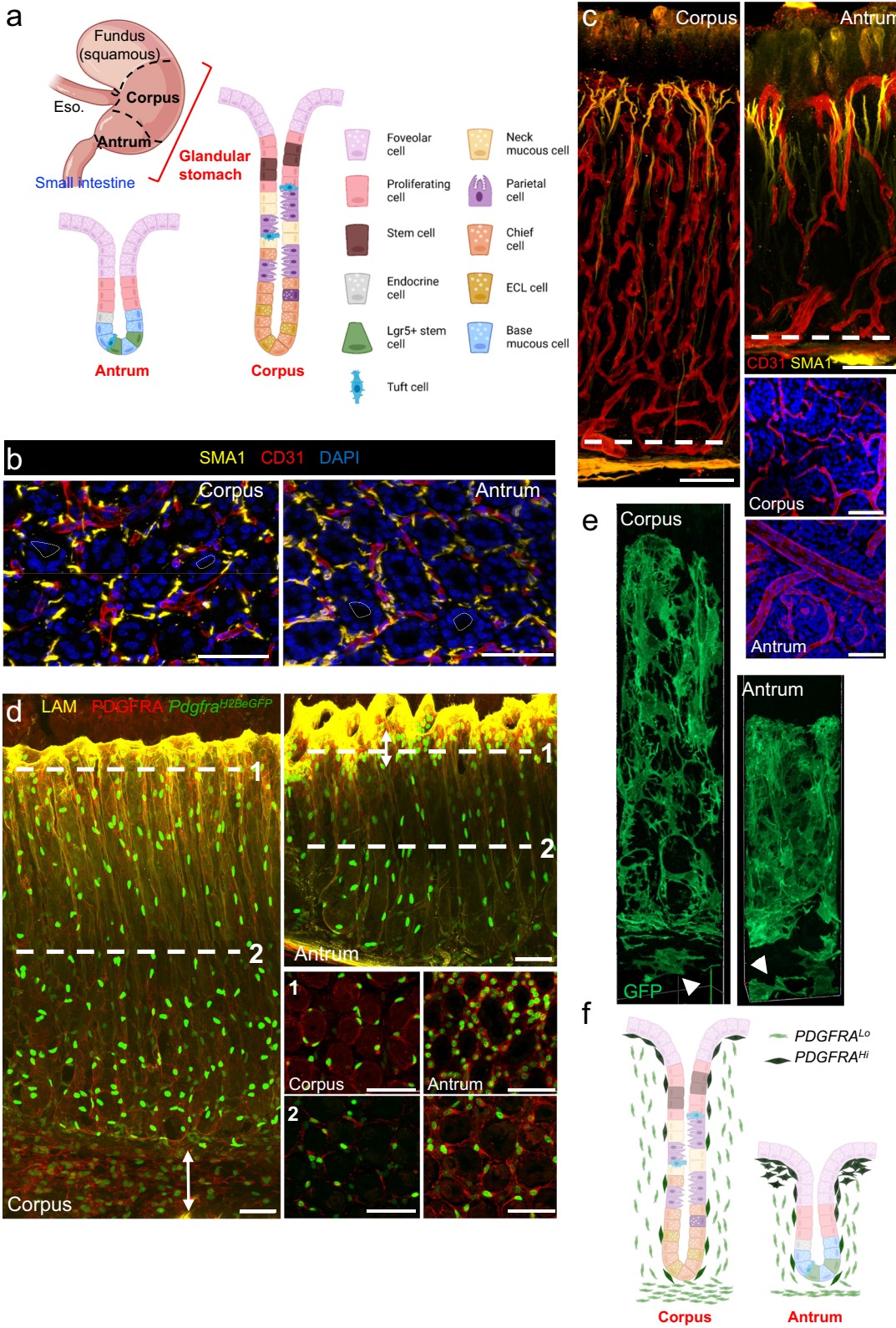

The corpus epithelium, which turns over slower than antral or intestinal epithelia, also contains surface foveolar cells; the neck of each gland carries mucous neck and acid-secreting parietal cells, while the gland base houses zymogenic "chief" cells (molecular markers we use to identify cell types in this study are listed in Supplementary Fig. 1a). The isthmus of corpus glands, populated by presumptive stem cells and their transit-amplifying daughters, lies between the pit and the neck (Fig. 1a); progeny of these replicating cells migrate bidirectionally, pit-cell precursors toward the lumen and other precursors toward the gland base[11–13]. Corpus isthmus stem cells replicate frequently and replenish, at a minimum, the pit and upper neck regions[14,15]. A distinct population with stem-cell properties resides

**Fig. 1 | Structure and organization of gastric corpus and antral mesenchyme. a** Stomach and intestinal regions and different organizations of glandular stomach epithelia. **b** Cross-sections of corpus (top) and antrum (bottom), showing smooth muscle actin (SMA1, yellow)-expressing smooth muscle fibers and CD31+ (red) capillaries packed tightly in the stroma between glands (nuclei stained with DAPI, blue). Dashed lines delineate gland lumen. Scale bars 50 μm. Representative images from three independent experiments. **c** Whole-mount 3D rendering of corpus (left) and antral (right) glands, highlighting structural differences in muscle fibers (SMA1, yellow) and capillaries (CD31, red) in the two regions. Tissue cross-sections at the level marked by dashed lines are shown in the bottom right (DAPI in blue, CD31 in red). Scale bars 50 μm. **d** Whole-mount 3D rendering of corpus (left) and antral (top right) glands in *Pdgfra*^H2BeGFP mice. Laminin (yellow) marks the basal lamina and highlights gland pits. Tissue cross-sections at the levels marked 1 and 2 are shown in the bottom right, with PDGFRA^Hi cells in neon green and PDGFRA^Lo cells in light green. Double-headed arrow highlights PDGFRA^Lo layer beneath corpus gland bottoms. Immunostaining with PDGFRA antibody (red) shows the extent of PDGFRA+ cytoplasm enveloping each gland. Scale bars 50 μm. **e** Single frames extracted from 3D video renderings of *Pdgfra*^Cre(ER-T2);*Rosa26*^mT/mG mouse (stained green after tamoxifen treatment) corpus (left) and antrum (right) to show the morphology of *Pdgfra*^Hi cells enveloping each gland and additional *Pdgfra*^Lo cells occupying space beneath glands (white arrowheads). Full videos of corpus and antral gland 3D renderings are included in the Supplementary information, see Supplementary Movies 5 and 6. **f** Schematic representation of the distributions of PDGFRA^Hi (dark green) and PDGFRA^Lo (light green) cells in relation to corpus and antral glands.

among chief cells at the gland base, expresses Wnt pathway genes *Troy* and *Lgr5*, and rarely replicates at rest but becomes activated upon injury to replenish cell losses in the basal and lower neck regions[16–21].

The balance that active and facultative gastric stem cells strike between self-renewal and differentiation reflects the epithelial response to signals from underlying mesenchyme. Active antral stem cells, for example, require Wnt signaling to maintain their state and proliferate[3,7,22], and facultative corpus stem cells are recruited to replicate in part by Wnt signaling[16,17]. Conversely, bone morphogenetic protein (BMP) signaling drives epithelial differentiation[23–25] and inhibitors of the pathway (BMPi) are implicated in gastric metaplasia and *Helicobacter pylori* infection[26,27]. Whereas recent studies have defined anatomic and functional relations between intestinal stromal and epithelial compartments in increasing detail[28–36], gastric mesenchyme is as yet superficially characterized. By combining corpus and antral cells, a recent single-cell analysis obscured possible differences between regionally distinct populations[37]. Here we report on our structural, molecular, and functional characterization of whole tissues and single cells from adult mouse corpus and antral mesenchyme. We define notable similarities and differences between epithelial and underlying stromal elements in the corpus, antrum, and small intestine (SI). Focusing on cells that express PDGFRA, we identify sub-glandular CD55-expressing fibroblasts as functional counterparts of intestinal trophocytes in the stomach.

## Results

### Anatomic definition and signaling potential of antrum and corpus mesenchyme

Foveolar cells are present in both gastric regions; their fraction is larger in the antrum, where pits (foveolae, outlined by staining with the lectin UEAI or with laminin antibodies) are wider and deeper in relation to the whole gland than those in the corpus (Supplementary Fig. 1b, c). In both regions, whole-mount confocal imaging revealed blood vessels and smooth muscle fibers packed tightly in the limited space between individual glands (Fig. 1b). Corpus arterioles branch along the full gland length to extend a deep capillary network, while antral arterioles have a larger bore and extend capillaries mainly at gland tops, with little branching along the gland length (Fig. 1c and Supplementary Fig. 1d, e, Supplementary Movies 1 and 2). Slender smooth muscle fibers extend lateral branches near the foveolar base, so the resulting fine network of lamina propria myocytes is taller and wider in the antrum (Fig. 1c and Supplementary Movies 3 and 4).

*Pdgfra*-expressing mesenchymal cells are pivotal in intestinal epithelial homeostasis[33]. In *Pdgfra*^H2BeGFP knock-in mice[38,39], we detected cells with high and low levels of nuclear GFP (Fig. 1d and Supplementary Fig. 1e–g). Antral PDGFRA^Hi cells aggregate around the pit, where they extend beyond the basement membrane into the lamina propria (Supplementary Fig. 1e, f), while corpus PDGFRA^Hi cells distribute uniformly along corpus glands (Supplementary Fig. 1g). PDGFRA^Lo cells distribute uniformly along the antral gland length but their density is modestly lower near the corpus isthmus and neck; in both gastric regions, PDGFRA^Lo cells also concentrate below the gland base, occupying a substantially larger space in the corpus than in the antrum (Fig. 1d and Supplementary Fig. 1g). To further assess cell morphology, we crossed *Rosa26*^mT/mG reporter[40] and *Pdgfra*^Cre(ER-T2) knock-in mice[41], thus producing mice where baseline red fluorescence (Tomato) in all cell membranes turns green (GFP) in cells with Cre enzyme activity (Supplementary Fig. 2a). Tamoxifen did not induce structural changes that indicate tissue toxicity (Supplementary Fig. 2a). Cells embedded in the basal lamina, corresponding to PDGFRA^Hi, engulf glands within their intercalated cytoplasmic projections and cells corresponding to PDGFRA^Lo are present beneath glands (Fig. 1e and Supplementary Movies 5 and 6). In summary, PDGFRA+ cells are present throughout the glandular stomach and their different arrangements in the corpus and antrum (Fig. 1f) likely reflect distinct influences on overlying gastric epithelia.

SI crypts form organoid structures in Matrigel supplemented with EGF, Noggin (a BMPi), and RSPO1, a Wnt potentiating agent (ENR medium)[42]. As *Lgr5*+ stem cells in the antral gland base resemble those in SI crypt bottoms, the original conditions to culture antral glands were adapted from an SI foundation and include additional requirements for Wnt3a and gastrin (ENRWG)[3]. Selected intestinal PDGFRA^Lo cells, which express BMP inhibitor (BMPi) and RSPO genes, induce intestinal crypts to form spheroids in vitro in the absence of any supplemental factors[33]. PDGFRA^Lo cells isolated by flow cytometry from either gastric segment (Supplementary Fig. 3a) failed to expand intestinal crypts in vitro, revealing a functional distinction from their intestinal counterparts (Fig. 2a). In contrast, corpus and antral glands readily formed spheroid structures in the absence of recombinant (r) factors when co-cultured with PDGFRA^Lo cells isolated from the SI or from the corresponding gastric region (Fig. 2b) but not when co-cultured with gastric stromal PDGFRA^neg or CD31+ endothelial cells (Supplementary Fig. 3b). Spheroids arising from corpus glands were consistently smaller and fewer than those from antral glands but expressed corpus-specific (chief and parietal cell) markers, while antral spheroids expressed higher levels of Wnt target genes *Axin2* and *Cd44* (Supplementary Fig. 3c). These structures thus reflect region-specific cell differentiation, different degrees of corpus and antral progenitor cell expansion, and discernible effects from mesenchymal cell co-cultures. Because isolated gastric PDGFRA^neg cells showed negligible activity in the assay, we focused on characterizing PDGFRA+ cells further.

### Regionality of gastrointestinal Pdgfra-expressing mesenchymal cells

To define regional differences, first we profiled bulk transcriptomes of PDGFRA+ cells purified by flow cytometry from the corpus and antrum of *Pdgfra*^H2BeGFP mice. Stripping external muscle before cell isolation excluded PDGFRA^Hi neurons nestled therein, so that the remaining PDGFRA^Hi cells represent gastric sub-epithelial myofibroblasts (SEMFs). Replicate RNA-seq libraries from each cell isolation were highly concordant (Supplementary Fig. 3d). In unsupervised

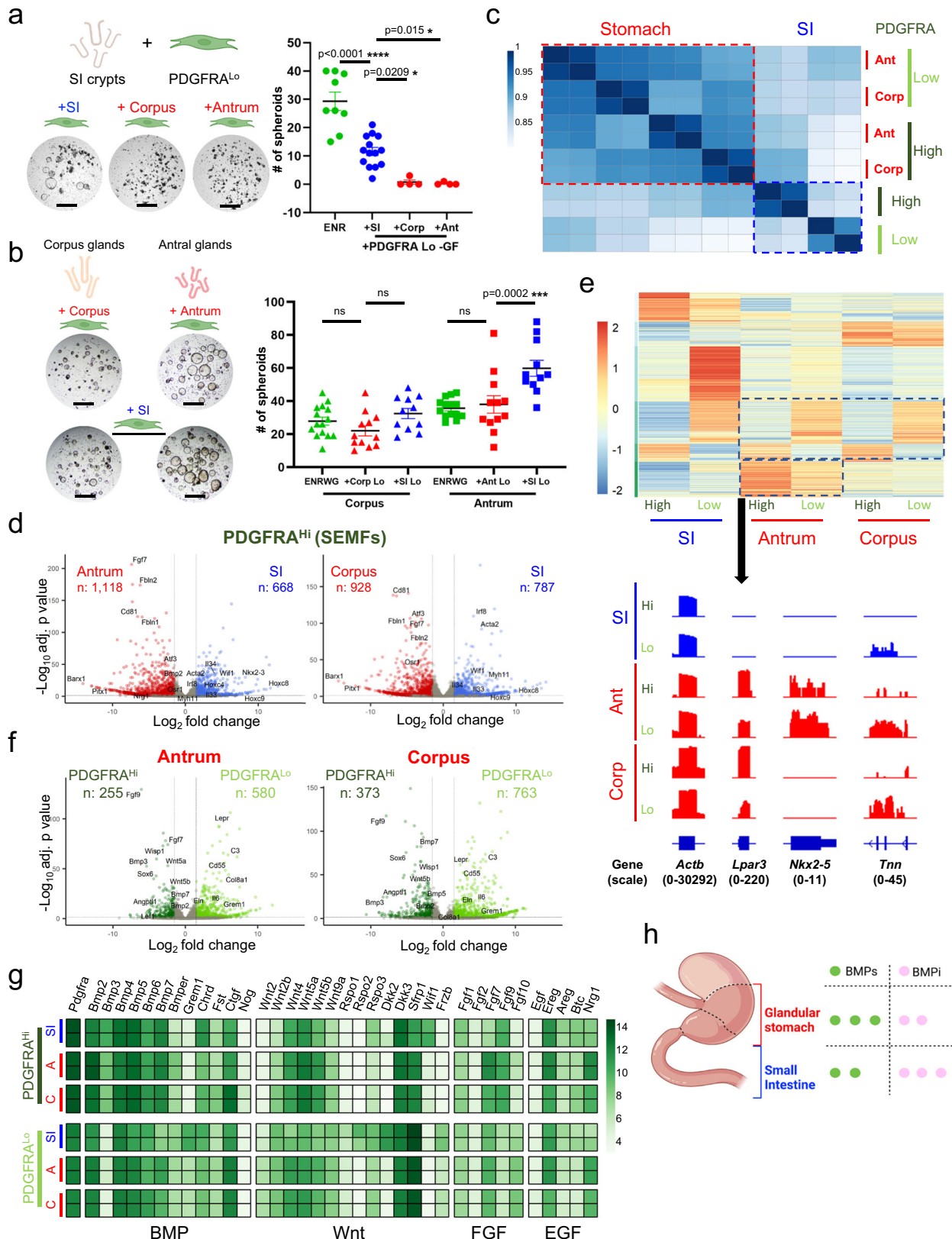

hierarchical analysis, gastric PDGFRA+ cells were distinct from their previously described intestinal counterparts[33], followed by differences between PDGFRA[Hi] and PDGFRA[Lo] cells in each organ (Fig. 2c). Corpus and antral PDGFRA[Hi] cell (SEMF) transcriptomes resembled each other more than they resembled SEMFs from the SI (Fig. 2d, Supplementary Fig. 3e and Supplementary Data 2). Exclusive expression of known

transcriptional regulators *Barx1* and *Pitx1* in gastric, and of *Nkx2-3* and posterior *Hox* genes in intestinal, cells support the veracity of additional findings. For example, gastric SEMFs express higher levels of *Fgf7*, *Fbln1* and *Fbln2*, *Cd81*, and transcription factor genes *Atf3* and *Osr1*, whereas intestinal SEMFs express higher levels of *Il33* and *Il34*, the Wnt inhibitor *Wif1*, and transcription factor *Irf8*. Differential expression

**Fig. 2 | Molecular distinction of corpus and antral PDGFRA-expressing mesenchyme. a** Co-culture of SI crypts with PDGFRA$^{Lo}$ cells isolated from SI, antral or corpus mesenchyme and without NOG or RSPO1 (−GF). Stomach PDGFRA$^{Lo}$ cells did not induce spheroids. SI organoid growth in complete ENR served as a control. Scale bars 400 μm. Bars represent mean ± SEM. ENR: $n = 9$, +SI $n = 14$, +Corp and +Ant: $n = 4$. Significance of differences determined by one-way ANOVA and Tukey's multiple comparison test. ns not significant. Source data provided as a Source Data file. **b** Co-culture of corpus or antral glands with PDGFRA$^{Lo}$ cells isolated from the same segment or from SI elicited spheroid growth in the absence of rNOG and RSPO1. Scale bars 400 μm. Corpus: ENRWG $n = 15$, +Corp Lo $n = 12$, +SI Lo $n = 11$, Antrum: ENRWG $n = 14$, +Ant Lo $n = 12$, +SI Lo $n = 13$. Control: gastric spheroid growth in ENRWG medium. Bars represent mean ± SEM. Significance of differences determined by one-way ANOVA and Sidak's multiple comparison test. ns not significant. Source data provided as a Source Data file. **c** Pearson correlations among RNA-seq libraries from SI, antral, and corpus PDGFRA$^{Hi}$ and PDGFRA$^{Lo}$ cells isolated from *Pdgfra*$^{H2BeGFP}$ mice. Differences between gastric and intestinal PDGFRA$^+$ cells are detailed in Supplementary Data 1. **d** Genes differentially expressed ($q < 0.05$; log$_2$ fold-difference >1.5) between SI, antral, and corpus PDGFRA$^{Hi}$ cells. Selected genes are named. Significance of differences determined by DESeq2 default settings (Wald test). **e** Genes enriched in each regional population ($q < 0.05$; log$_2$ fold-difference >1.5; $k$-means = 6) and Integrative Genome Viewer tracks for representative genes enriched in antral PDGFRA$^+$ cells. *Actb* controls for normalization of read counts. Parentheses: range of signal values. **f** Genes differentially expressed ($q < 0.05$; log$_2$ fold-difference >1.5) between antral and corpus PDGFRA$^{Hi}$ and PDGFRA$^{Lo}$ cells. Significance of differences determined by DESeq2 default settings (Wald test). **g** Relative expression of mRNAs for candidate niche factors (BMP, Wnt, FGF, and EGF agonists and antagonists) in each mesenchymal cell population. Heatmap is prepared from normalized rlog counts from DEseq2; each row represents one replicate. **h** Bulk RNA-seq data point to a higher BMP tone from *Pdgfra*$^+$ cells in stomach than in the SI.

of *Acta2* and *Myh11* hints that intestinal and antral SEMFs may be more contractile than those in the corpus.

Segment- and cell type-specific gene modules were evident in each population (Fig. 2e). Intestinal PDGFRA$^{Lo}$ cells have the most distinctive transcriptome of any population, with unique expression of *Eln*, *Col8a1*, and *Il6*, among other factors, and differing from intestinal SEMFs in expression of >2100 genes ($q < 0.05$; |log$_2$ fold-change| > 1.5), compatible with distinct and opposing functions[33] (Supplementary Fig. 3f). The corresponding populations in both corpus and antrum are less divergent, differing in expression of ~800 (antrum) to ~1100 (corpus) genes (Fig. 2f). Gastric PDGFRA$^{Lo}$ cells from either segment are enriched for *Cd55* (a marker we investigate later in this study), *Lepr* and *Dll1* (genes that mark specific mesenchymal cells in mouse and human intestinal mesenchyme, respectively)[43,44] while gastric SEMFs are enriched for signaling factors such as *Fgf9* and *Angptl1*, multiple BMP genes, and for transcription factor (TF) gene *Sox6* and, in the antrum, Wnt-responsive *Lef1* (Fig. 2d, e and Supplementary Data 2). *Foxl1*, which marks intestinal SEMFs among other cells[31,33,45], is expressed in gastric SEMFs and, at a lower level, also in gastric PDGFRA$^{Lo}$ cells. Thus, regional differences in the spatial arrangements of *Pdgfra*-expressing cells (Fig. 1) extend to their gene expression profiles.

Among potential sources of the signals that support epithelial expansion in vitro, genes for canonical *Wnt2* and for non-canonical ligands *Wnt4*, *Wnt5a*, and *Wnt5b* are expressed in both PDGFRA$^+$ populations in both corpus and antrum; however, Wnt-potentiating factors *Rspo1*, *Rspo2*, and *Rspo3* express at lower levels in gastric than in intestinal PDGFRA$^{Lo}$ cells (Fig. 2g). Thus, while PDGFRA$^+$ cells in every segment produce Wnt ligands, only transcripts in intestinal PDGFRA$^{Lo}$ cells display a potential for significant canonical Wnt signaling. BMP transcripts are enriched not only in intestinal, antral, and corpus PDGFRA$^{Hi}$ cells, but *Bmp2*, *Bmp5*, and *Bmp7* are also prominent in gastric PDGFRA$^{Lo}$ cells (Fig. 2g). *Grem1*, a BMPi abundant in intestinal PDGFRA$^{Lo}$ cells, and other classical BMPi are barely present in bulk populations of gastric PDGFRA$^{Lo}$ cells, which instead express a putative and atypical BMP7 inhibitor, *Ctgf*[46,47]. Based on mRNA levels alone, these findings suggest that overall BMP "tone" may be higher in the stomach than in the SI (Fig. 2h).

In culture, gastric glands generally form spheroid structures (see Fig. 2b and Supplementary Fig. 3b), distinct from intestinal organoids, in which budding protrusions represent crypts housing progenitor and stem cells[42]. When conditions to culture gastric glands were first defined, human fibroblast growth factor (FGF) 10 induced budding of unclear significance (stomach epithelium lacks crypts)[3]. In this light, the asymmetry of FGF transcripts in PDGFRA$^+$ cells is striking. Of the five transcripts expressed in any population, *Fgf9* is expressed in SEMFs and *Fgf10* in PDGFRA$^{Lo}$ cells from all sites, while *Fgf7* is especially high in antral SEMFs. *Nrg1*, an EGF-family factor implicated in ISC regeneration after injury[34] and corpus pit cell differentiation[48], is also

uniquely higher in gastric (especially antral) than in intestinal SEMFs and PDGFRA$^{Lo}$ cells (Fig. 2g).

## Transcriptional features of antral and corpus mesenchyme

Having identified differences in sub-epithelial structure and signaling capacity, we examined cellular heterogeneity by mRNA profiling of whole gastric mesenchyme (manually depleted of external smooth muscle) at single-cell resolution. A total of 34,445 mesenchymal cells (20,624 from corpus and 13,821 from antrum; epithelial contamination was negligible) provided information on at least 900 transcripts each (2700 transcripts for antrum Rep.1, Supplementary Fig. 4a). Consistent with the structures and cells revealed by fluorescence microscopy, graph-based clustering identified substantial fractions of *Cd31*$^+$ endothelial cells from corpus, *Acta2*$^{hi}$ *Myh11*$^{hi}$ myocytes, *Acta2*$^{lo}$ *Myh11*$^{lo}$ *Rgs5*$^+$ pericytes, and PDGFRA$^+$ cells that lack these markers (Fig. 3a). Less than 10% of whole isolates were *Ptprc*$^+$ immune cells, smaller than the 45% fraction present in whole SI mesenchyme[33], only ~1% of cells were glial, and <1% showed signs of ongoing cell replication (Fig. 3a). Despite structural differences in corpus and antral blood vessels and muscle fibers (Fig. 1), transcriptomes of their constituent cells showed few regional differences. Significant cluster distinctions between corpus and antrum mapped exclusively to *Pdgfra*$^+$ cells (Fig. 3b and Supplementary Fig. 4b), which express more candidate niche factors than other mesenchyme (Supplementary Fig. 4c). As all other cell types from corpus and antrum resembled each other similar *Pdgfra*$^+$ cell dissimilarities are unlikely to represent a batch effect, as we confirmed additionally with AUGUR analysis[49] (Supplementary Fig. 4d). These findings justify our preceding attention and subsequent focus on the 5896 antral and 6594 corpus *Pdgfra*$^+$ cells from our scRNA-seq study.

Graph-based analysis limited to these cells identified 6 clusters: *Pdgfra*$^{Hi}$ and two related but distinct *Pdgfra*$^{Lo}$ cell populations from each gastric segment (Fig. 3c and Supplementary Fig. 2b—note that *Cd34* marks intestinal *Pdgfra*$^{Lo}$ but not *Pdgfra*$^{Hi}$ cells[33]). Two *Pdgfra*$^{Lo}$ cell types in the corpus, CorpLo and CorpLo1, were separated on UMAP graphs but, consistent with bulk RNA profiles (Fig. 2e), antral cells separated less well. UMAP distinguished antral *Pdgfra*$^{Hi}$ (AntHi) and *Pdgfra*$^{Lo}$ (AntLo) cells and a substantial (44% of *Pdgfra*$^+$ cells) third pool, AntInt, with intermediate *Pdgfra* and *Cd34* levels and a distinctive RNA profile (Fig. 3c, d); this profile includes enriched expression of the transcription factor (TF) gene *Sox11*, contractile genes *Acta2* and *Myh11*, and prostaglandin synthase *Ptgs2* (Fig. 3e and Supplementary Data 3). Some features resemble AntHi (e.g., EGF-family genes *Ereg*, *Areg* and *Nrg1*, the TGF-ß inhibitor *Tgfbi*, and *Col4a5*) and others resemble AntLo (e.g., *Tgfb2*, Wnt antagonist *Sfrp1*, and extracellular matrix factors *Spon2*, *Col15a1*, *Fbln1* and *Fbln2*). Notably, *Bmp2/5/7* and non-canonical *Wnt4* transcript levels approach those in antral and intestinal SEMFs (*Pdgfra*$^{Hi}$ cells), while other signals (e.g., *Wnt5a*, *Fgf7*) are lower than in SEMFs (Fig. 3e, f and Supplementary Fig. 4c, e).

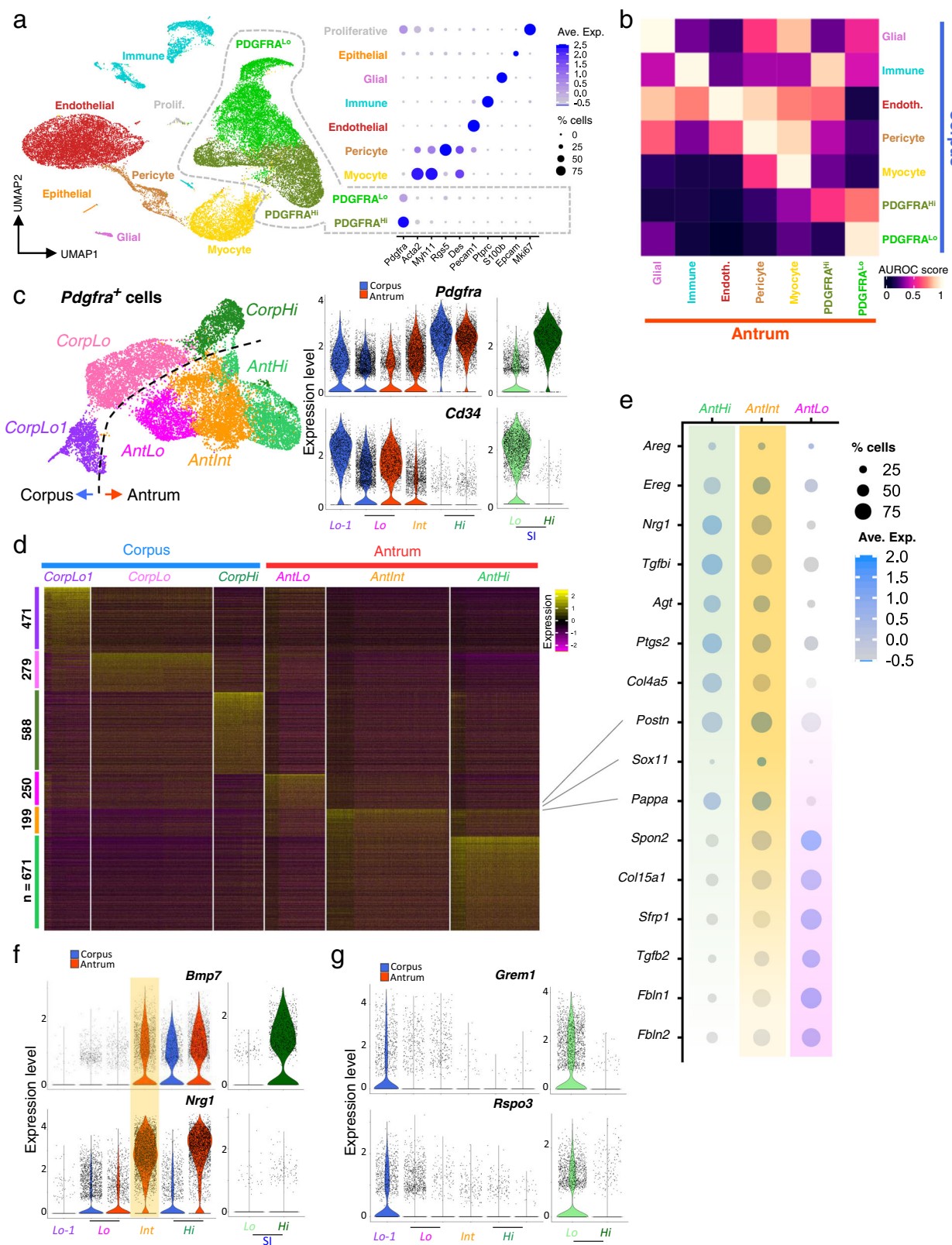

Because flow cytometry separated PDGFRA[Hi] and PDGFRA[Lo] cells from *Pdgfra*[H2BeGFP] mouse corpus better than the antrum (Supplementary Fig. 3a), we infer that AntInt segregates between these two populations in FACS isolation. The above differences, however, mark them as distinct cells and we show below that they differ from AntLo in niche activity.

Differences between corpus and antral *Pdgfra*[Lo] subpopulations also include expression of known niche factors. Low levels of intestinal niche factors *Grem1* and *Rspo3* apparent in bulk RNA profiles of stomach PDGFRA[Lo] cells (Fig. 2g) are largely limited to the distinct CorpLo1 cluster and AntLo, albeit at levels lower than intestinal *Pdgfra*[Lo] cells (Fig. 3g). CorpLo1 also expresses a different spectrum of

**Fig. 3 | Delineation of corpus and antral PDGFRA⁺ mesenchymal cell populations. a** Left: Uniform manifold approximation and projection (UMAP) plot of mRNA profiles from 34,445 single cells isolated from corpus (20,624 cells) and antral (13,821 cells) mesenchyme. Right: Relative expression of known cell-restricted genes in the UMAP cell clusters. Circle diameters represent the cell fraction expressing a gene and shades of purple represent normalized average expression within the population. **b** MetaNeighbour AUROC (area under the receiver operating characteristic) analysis shows high similarity between corresponding antral and corpus mesenchymal cells of each type other than the PDGFRA^Hi and PDGFRA^Lo cell populations. **c** Left: UMAP identification of distinct populations of *Pdgfra*-expressing cells extracted from scRNA analysis of unfractionated mesenchyme. Right: Relative expression of *Pdgfra* and *Cd34* in the identified *Pdgfra*-expressing populations and the corresponding SI cells. Blue, corpus; red, antrum; green, SI. The 3 corpus populations are more distinct than the 3 antral populations. **d** Differential genes expressed among the six PDGFRA⁺ mesenchymal cell types (Wilcoxon Rank Sum test, logfc.treshold = 0.25). The genes are listed in Supplementary Data 2. **e** Relative expression of marker genes from the three antral PDGFRA⁺ subpopulations identified by scRNA-seq. Circle sizes represent the percentage of cells expressing a gene, fill colors represent normalized average expression. AntInt shows distinct intermediate patterns. **f** Expression of niche genes *Bmp7* and *Nrg1* in each population compared with *Pdgfra*-expressing SI mesenchymal cells. AntInt expression levels are comparable to those in antral *Pdgfra^Hi* cells (AntHi). *Nrg1* expression in corpus and SI is significantly lower than in antral cells. **g** *Grem1* and *Rspo3* expression in each population compared with *Pdgfra*-expressing SI cells. Both genes are restricted to *Pdgfra^Lo* populations and expressed at lower levels in the stomach than in SI.

extracellular matrix genes (high *Has1*, *Dpt* and *Col14a1*—Supplementary Data 3) compared to CorpLo, suggesting that these cells may localize in different anatomic domains. Of note, BMP agonist mRNA levels diverge less between *Pdgfra^Hi* and other *Pdgfra^+* cells in the stomach than in the intestine and BMPi express at lower levels (Supplementary Fig. 4f), affirming ostensibly higher BMP tone in the stomach. MetaNeighbor showed that AntInt most resembles AntLo (Supplementary Fig. 4g). In summary, we identify distinct gastric PDGFRA⁺ populations from each region and below we address their properties systematically.

## Molecular and spatial heterogeneity of antral SEMFs

We first considered *Pdgfra^Hi* cells (SEMFs), which occupy a specific anatomic compartment and have unique mRNA profiles (Figs. 1–3). Gastric and intestinal SEMFs differ in high *Fgf7* (antrum) and *Fgf9* (corpus) expression and in profiles of the TF genes *Foxl1* and *Gli1* (Fig. 2g and Supplementary Figs. 4c and 5a). Considered apart from other mesenchymal cells, they divided into one corpus and two antral subpopulations, all of which express canonical BMP ligand genes (Fig. 4a and Supplementary Fig. 5b). Notably, one antral cluster expresses high *Fgf7* and the EGF ligand *Nrg1*, while the other is marked by high *Bmp3* and *Ctgf* (candidate non-canonical BMPi)[50,51] and low levels of *Nrg1*, a profile similar to corpus SEMFs (Fig. 4a, b); *Fgf7* and *Bmp3* are largely SEMF-restricted, while the other genes are enriched in SEMFs but also expressed in other PDGFRA⁺ cells (Supplementary Fig. 4c, f). In situ hybridization localized antral *Fgf7⁺ Nrg1⁺* SEMFs to the tops of antral glands, adjacent to foveolae, whereas *Bmp3* and *Ctgf* localize primarily in SEMFs that abut the zone of epithelial cell replication in the lower gland (Fig. 4c). As antral SEMF density is higher at gland tops than at the base (Supplementary Fig. 1e–g), the proportions of SEMF subpopulations (Fig. 4a) further support that molecularly distinct antral SEMF subpopulations are spatially segregated.

Because *Bmp3* and *Ctgf* may have BMPi activity[50,51] and SEMFs expressing those genes lie near proliferative epithelial cells, we asked if they might support antral gland growth in vitro; to optimize the assay for BMPi activity, we reduced RSPO and Wnt3 concentrations 5-fold. In these conditions, rNOG readily supported antral spheroids, but the same concentration of human rBMP3 or rat rCTGF lacked that activity (Supplementary Fig. 5c). No surface marker reliably separated *Bmp3⁺ Ctgf⁺* from *Fgf7⁺ Nrg1⁺* SEMFs. When co-cultured with gastric glands in the absence of any recombinant factor, unfractionated FACS-sorted antral or corpus PDGFRA^Hi cells (GFP^Hi cells from *Pdgfra^H2BeGFP* mice) did not induce spheroid growth from the corresponding glands, in contrast to the stimulation by ENRWG medium (Fig. 4d) or by bulk PDGFRA^Lo cells (Fig. 2b). Together, these findings reveal spatial and molecular asymmetry of antral SEMFs that fail to support corpus or antral gland growth in vitro, much as intestinal SEMFs lack discernible support function in organoid assays[33]. To better define SEMF contributions in those assays, we co-cultured gastric glands with SEMFs in complete ENRWG medium, in medium lacking EGF and Noggin, or in medium containing RSPO1 as the only supplement. In the presence of

these factor combinations, SEMFs did not reduce the number of resulting spheroids (Supplementary Fig. 5d).

In cultured antral glands, recombinant human FGF10 induces buds, which are presumed to reflect epithelial maturation[3]. Because *Fgf7* is abundant in peri-foveolar antral SEMFs, belongs in the same clade as *Fgf10*, and their common receptor, *Fgfr2*, is expressed in antral epithelium[52], we considered a function for FGF7 in epithelial differentiation. Addition of murine rFGF7 to ENRWG medium increased budding substantially without affecting organoid numbers (Fig. 4e). Although constitutive *Fgf7* gene disruption has no reported effects on mouse gastric development or function[53] subtle differentiation defects could have been masked. To assess possible gastric functions in adult mice, we used CRISPR gene editing to flank exon 2 of the *Fgf7* locus with LoxP sites (Supplementary Fig. 6a), then introduced the floxed allele onto the *Pdgfra^Cre(ER-T2)* background[41]. *Pdgfra^Cre(ER-T2)*;*Rosa26^LSL-tdTomato* mouse stomachs revealed robust Cre recombinase activity in PDGFRA^Hi and, at lower efficiency, in PDGFRA^Lo cells (Supplementary Fig. 6a, see also Fig. 1e and Supplementary Movies 5 and 6). PCR genotyping verified the expected null allele in Tomato⁺ cells isolated from antral *Pdgfra^Cre(ER-T2)*;*Fgf7^Fl/Fl*;*Rosa26^LSL-TdTomato* mesenchyme (Supplementary Fig. 6a). We observed no overt histologic defects or perturbed distribution of antral epithelial proliferative or differentiation markers (Supplementary Fig. 6b) and scRNA profiles of purified wild-type (WT) and *Fgf7⁻/⁻* antral epithelium showed intact proportions and expression characteristics of proliferative and differentiated cells (Supplementary Fig. 6c), with only subtle differences in foveolar cell transcripts between *Fgf7⁻/⁻* and WT mice (Supplementary Data 3). Thus, despite the abundance and specificity of *Fgf7* expression in a spatially discrete subpopulation of antral SEMFs and its budding effect in vitro, its essential, non-redundant role in maintaining antral epithelium in vivo is at most minor.

## Spatially and molecularly distinct corpus and antral PDGFRA^Lo cell populations

Next, we examined gastric *Pdgfra⁺* cell clusters other than *Pdgfra^Hi* (Fig. 3). Considered separately from SEMFs, *Pdgfra^Lo* and *Pdgfra^Int* resolved into 5 sub-groups within 3 clusters; the smallest and most distinct cluster corresponds to CorpLo1 and few antral *Pdgfra^Lo* cells, while the remaining cells cluster by corpus and antral origins (Fig. 5a). Within the large corpus cluster, one sub-group, CorpLo2, shares expression of a panel of niche factor genes with AntLo, while the other sub-group, CorpLo3, expresses genes (e.g., *Pappa*, *Sox11*, *Postn*, *Nrg1*— see Fig. 3e) that mark AntInt (Fig. 5a). These populations selectively express BMPs, which inhibit ISCs, and non-canonical Wnts, which lack known niche function (Fig. 5a, b). Conversely, factors known to constitute the ISC niche, such as *Grem1* and *Rspo3*, are selectively enriched in AntLo, CorpLo2, and the isolated corpus cluster CorpLo1 (Fig. 5a, b). Thus, scRNA data separate gastric *Pdgfra^Lo* cells into fractions that superficially resemble cells active in the ISC niche (AntLo, CorpLo1, CorpLo2) and those that express distinct, potentially opposing factors (AntInt and CorpLo3).

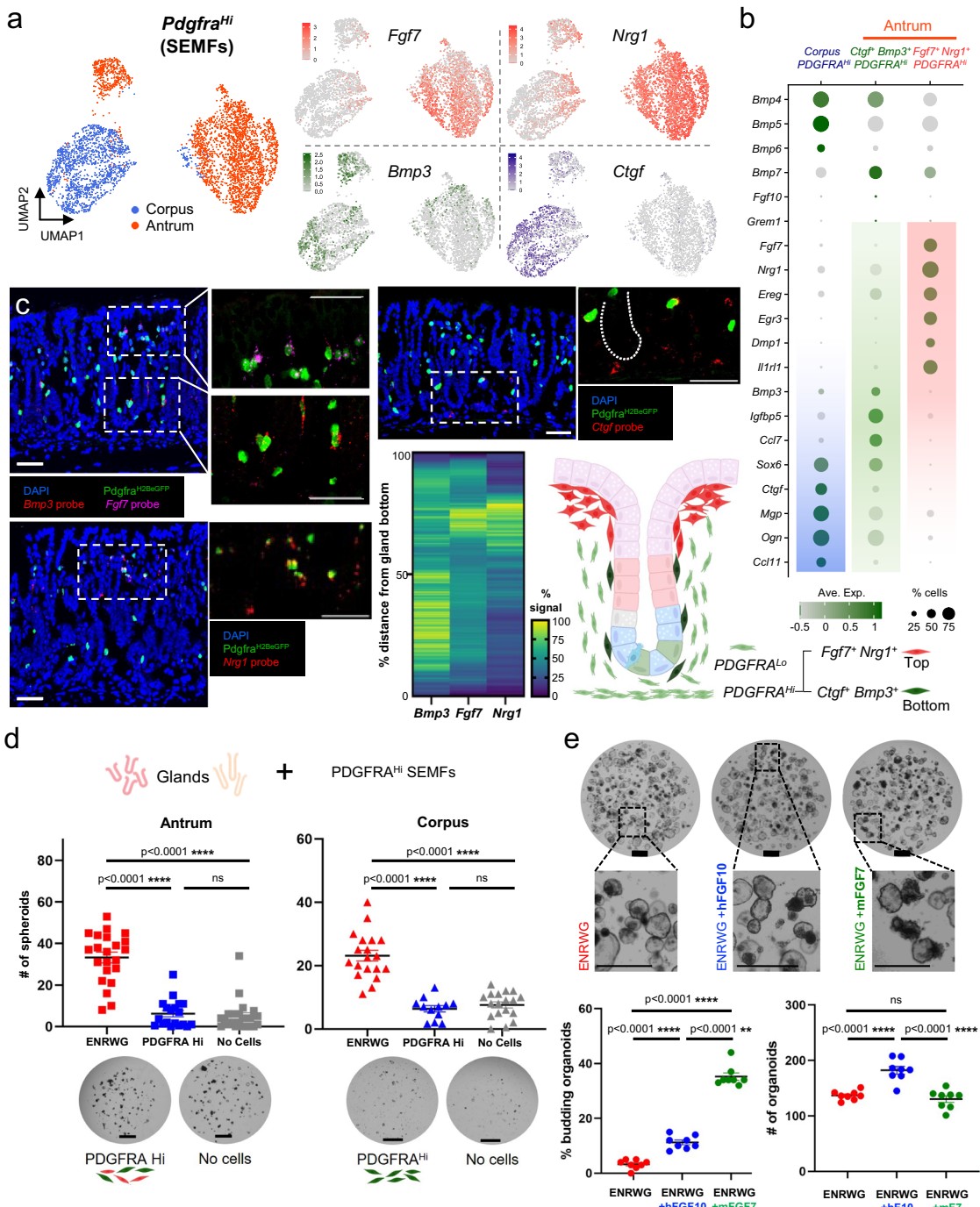

**Fig. 4 | Identification and functional assessment of distinct gastric SEMF populations. a** In isolation, corpus and antral SEMFs are distinct and antral SEMFs divide further into two cell populations. Right: *Fgf7, Nrg1, Bmp3,* and *Ctgf* mRNA densities projected on the UMAP plot. **b** Relative expression of scRNA-seq markers from the three SEMF populations. Circle diameters and fill colors represent the fraction of cells expressing a gene and normalized average expression levels. The two antral SEMF populations differ in expression of *Fgf7, Nrg1, Bmp3,* and other factors; corpus SEMFs resemble the smaller antral *Bmp3*+ cluster more than the larger population of antral *Fgf7*+ *Nrg1*+ cells. **c** Fluorescence in situ hybridization (RNAscope) on antral tissue sections localizes *Fgf7*- and *Nrg1*-expressing PDGFRA^Hi SEMFs near gland pits and *Bmp3*- and *Ctgf*-expressing cells in the lower half of glands. Images represent fields examined in three independent experiments with each probe. Scale bars 50 μm. Heatmap: average fluorescence signal quantified along 16−25 individual glands (*Ctgf* probe quantification is shown in Fig. 6a). Source data are provided as a Source Data file. Right: schematic representation of antral

SEMF distribution. **d** Co-culture of antral and corpus glands with unfractionated PDGFRA^Hi cells, which fail to induce spheroids in the absence of rNOG and RSPO1. Glands cultured in ENRWG medium serve as controls. Scale bars 400 μm. Bars represent mean ± SEM. Antrum: +ENRWG *n* = 22, +PDGFRA^Hi *n* = 19, No Cells *n* = 22; Corpus: +ENRWG *n* = 19, +PDGFRA^Hi *n* = 12, No Cells *n* = 18. *n*: number of culture wells analyzed over six independent experiments. Significance of differences determined by one-way ANOVA coupled with Sidak's multiple comparison test. ****$p < 0.0001$, ns not significant. Source data are provided as a Source Data file. **e** Antral glands exposed to recombinant rFGF7 or rFGF10 in addition to complete ENRWG medium. Budding of spheroid structures increased upon treatment with rFGF7 without affecting spheroid numbers (*n* = 8 independent experiments). Bars represent mean ± SEM. Significance of differences determined by one-way ANOVA coupled with Dunnett's multiple comparison test. ns not significant. Scale bars 400 μm. Source data are provided as a Source Data file.

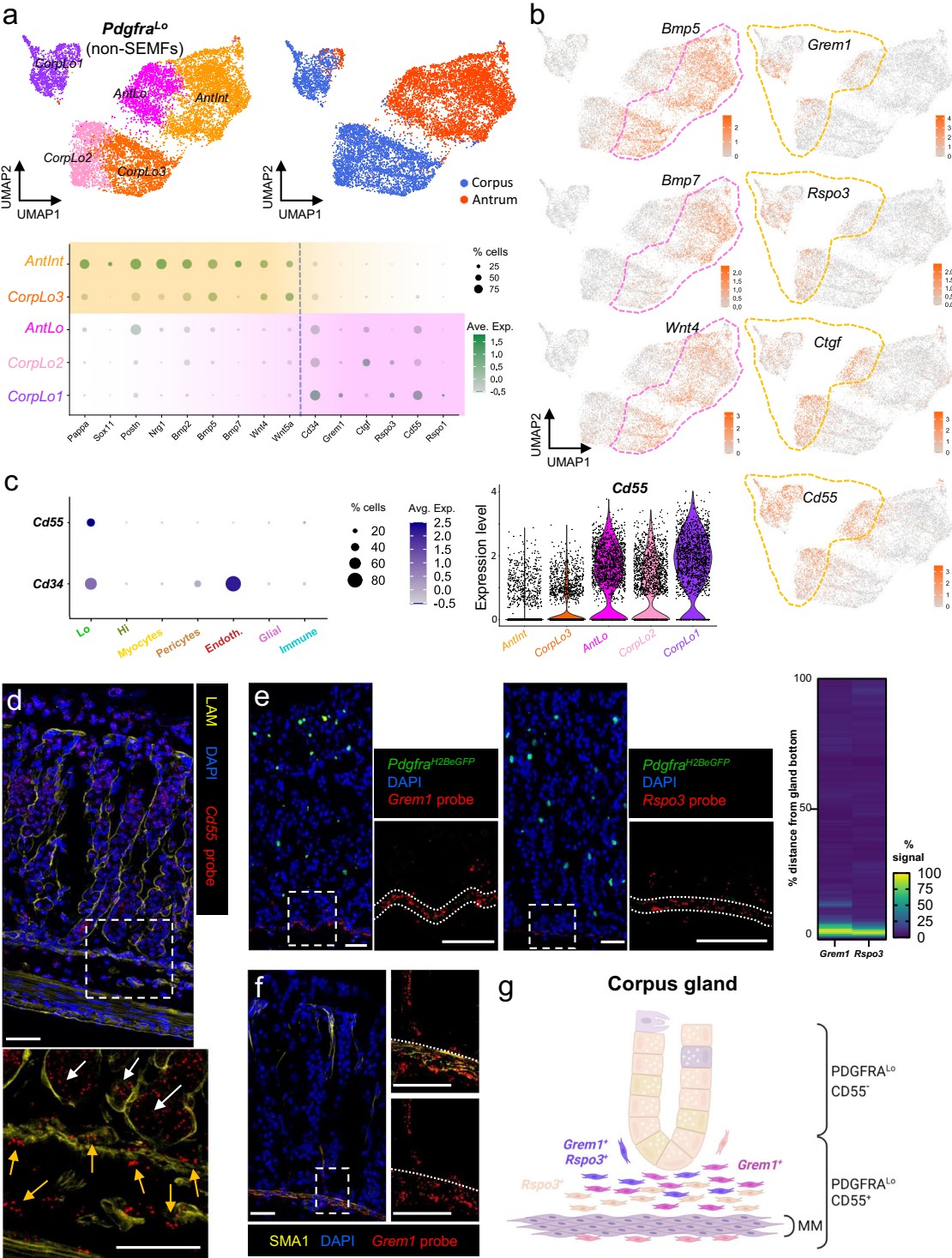

To assess niche cell functions in isolated gastric cell populations, we sought a cell surface marker that could separate *Pdgfra^Lo* cell fractions expressing *Grem1*, *Rspo3* or *Ctgf* (to varying degrees—shaded pink in Fig. 5a) from those expressing BMP and non-canonical Wnts (shaded orange). In scRNA analysis, two cell surface marker genes, *Cd34* and *Cd55*, showed selective expression in AntLo, CorpLo2, and especially CorpLo1 cells (Fig. 5a). *Cd34*, a known marker of trophocytes and other intestinal PDGFRA^Lo^ cells, is also abundant in endothelial cells[33,54] (Fig. 5c). In contrast, *Cd55*, which encodes the tetraspanin CD55 and marks SI mesenchymal cells that support in vitro expansion of SI crypts[35], is additionally expressed only in a small fraction of

gastric immune cells (at levels lower than in *Pdgfra^Lo* cells—Fig. 5c) and in the foveolar epithelium (Supplementary Fig. 7a). Flow cytometry of epithelial (EpCAM^+^, Supplementary Fig. 7b) and PDGFRA^neg^ non-immune (CD45^−^/PTPRC^−^) cells from *Pdgfra^H2BeGFP^* mice showed negligible surface CD55 expression, but reliably purified corpus and antral CD55^+^ PDGFRA^Lo^ cells (Supplementary Fig. 7c, d) and traced CD55 expression in CD45^+^/PTPRC^+^ cells mostly to B lymphocytes (Supplementary Fig. 7e). In fact, ~90% of B lymphocytes from each region expressed CD55 against 60% of myeloid cells and 70% of T lymphocytes. Whereas immunohistochemistry with multiple CD55 antibodies gave high background signals, in situ hybridization (ISH) with *Cd55*

**Fig. 5 | Identification and localization of *Cd55*-expressing gastric PDGFRA^(Lo) cells. a** In isolation, gastric PDGFRA⁺ mesenchyme other than SEMFs resolves into five subpopulations: previously identified AntInt, AntLo and CorpLo1 (Fig. 3) and two CorpLo subpopulations, CorpLo2 and CorpLo3. Relative expression of selected scRNA-seq markers is plotted below. Circle diameters: cell fraction expressing a gene, fill shades: normalized average expression. CorpLo3 shares markers identified in AntInt (see Fig. 3e). **b** *Bmp5, Bmp7, Wnt4, Grem1, Rspo3,* and *Ctgf* transcript densities projected on the UMAP plot of resolved non-SEMF cell clusters. CorpLo3 and AntInt express BMPs and *Wnt4*, while AntLo, Corp1, and Corp2 express *Grem1, Rspo3,* and *Ctgf*. Dashed lines mark the distinct populations. **c** Left: Relative *Cd55* and *Cd34* expression in sub-epithelial cell fractions from scRNA-seq analysis. Circle diameters and fill colors represent the parameters defined in (**a**). Middle: Relative *Cd55* expression in PDGFRA^(Lo) subpopulations. Right: Projection of *Cd55* density on the PDGFRA^(Lo) UMAP, showing nearly exclusive expression in CorpLo1, CorpLo2 and AntLo cells. **d** In situ hybridization (RNAscope)

of corpus tissue sections, together with LAM immunostaining, localizes *Cd55* in the epithelium (white arrows) and in PDGFRA^(Lo) mesenchymal cells (orange arrows) near the gland base, on both sides of muscularis mucosae. The area within the dashed box is magnified below. Images represent scores of fields examined in two independent experiments. Scale bars 50 μm. **e** In situ hybridization (RNAscope) of corpus tissue sections localizes *Grem1* (left) and *Rspo3* (middle) near the gland base. Images represent scores of fields examined in two independent experiments. Scale bars 50 μm. Dotted lines mark the muscularis mucosae. In the heatmap, fluorescence signal strength is quantified along 50–52 individual glands. Source data are provided as a Source Data file. **f** In situ hybridization (RNAscope) of corpus tissue localizes *Grem1* in the SMA1-immunostained muscularis mucosae, demarcated by dotted lines, and in the space corresponding to PDGFRA^(Lo) cells near the gland base (n = 2 independent experiments). Scale bars 50 μm. **g** Schematic illustration of the distribution of CD55⁺ PDGFRA^(Lo) cells relative to corpus glands and the muscularis mucosae (MM). *Grem1*- and *Rspo3*-expressing cells lie largely beneath glands.

probe showed that extra-epithelial *Cd55* expression is restricted to sub-glandular mesenchymal cells located below and just above the muscularis mucosae, into the bottom ~1/6 of corpus gland height (Fig. 5d).

To better define this mesenchymal *Cd55*⁺ population, we used ISH probes for *Grem1* and *Rspo3*, which scRNA-seq detected in CorpLo1 or in CorpLo1 and CorpLo2, respectively (Fig. 5a—high *Ctgf* ISH signals in corpus SEMFs occluded *Ctgf*-expressing PDGFRA^(Lo) cells). In *Pdgfra^(H2BeGFP)* mouse corpus, *Grem1* and *Rspo3* localized in adjacent sub-glandular mesenchymal cell layers (Fig. 5e). SMA1 (smooth muscle actin1, Fig. 5f) immunostaining placed the dense ISH signals in a tissue layer where PDGFRA^(Lo) cells intermingle with muscularis mucosae, while sparser signals localized to PDGFRA^(Lo) cells located between the muscularis and the corpus gland base (Fig. 5f). This distribution resembles that of *Cd55* but the latter extends farther below the muscularis mucosae than *Rspo3* and especially *Grem1* (Fig. 5d–f). Taken together with scRNA-seq, which showed broader expression of *Cd55* than the other markers (Fig. 5a), these findings indicate that CorpLo1 cells lie within and above the muscularis, CorpLo2 cells extend below the muscularis, and the largest subpopulation, CorpLo3 which lacks *Grem1* or *Rspo3*, surrounds the top and middle zones of corpus glands (Fig. 5g).

Extra-epithelial *Cd55* in the antrum also localized in sub-glandular mesenchyme (Fig. 6a). Consistent with low *Rspo3* expression in scRNA-seq data (Fig. 5a), ISH signals for *Rspo3* were weak, but *Grem1* and *Ctgf* localized confidently in sub-glandular mesenchyme, overlapping with *Cd55* (Fig. 6b, c): *Grem1*⁺ *Pdgfra^(Lo)* cells are mixed with muscularis mucosae and *Ctgf*⁺ *Pdgfra^(Lo)* cells reside between the muscularis and antral gland bottoms. Thus, *Cd55*⁺ *Pdgfra^(Lo)* cells are restricted to the space beneath both corpus and antral glands. ISH and scRNA-seq data were consistent in indicating that *Rspo3*, when present, is co-expressed with *Grem1*, but the overlap between *Ctgf* and *Grem1* is limited in both stomach regions (Supplementary Fig. 8a). The CD55⁺ fraction therefore represents a sub-glandular mix of *Grem1*⁺ *Rspo3*⁺ and *Grem1*⁻ *Ctgf*⁺ cells.

These findings highlight similarities and differences between corpus, antral, and SI *Pdgfra^(Lo)* mesenchyme. Antral *Pdgfra^(Lo)* cells express less *Grem1* or *Rspo3* than their SI counterparts (Fig. 2g), but the transcripts show substantively similar distribution in both sites, near proliferative and Lgr5⁺ stem cells at the base of antral glands and SI crypts (Fig. 6d). The apparent gradient of high to low *Grem1* and *Rspo3* expression (from muscularis to epithelium) in sub-glandular corpus *Pdgfra^(Lo)* cells also resembles a self-organized gradient recently described in the SI[35] (Fig. 5g). However, proliferative and presumptive stem cells in the isthmus of corpus glands lie far from *Grem1*- and *Rspo3*-expressing cells. Therefore, sub-glandular CD55⁺ cells may drive isthmus cell replication from a distance, may act principally on nearby facultative stem cells in the gland base, or may affect epithelial differentiation more than cell replication per se. BMP signaling, for example, is implicated not only in exit from the stem-cell

compartment[55–57] but also in intestinal endocrine[58] and gastric foveolar (pit) cell[25] differentiation.

## PDGFRA^(Lo) cell populations that affect epithelial growth in vitro and differentiation in vivo

Gastric mesenchymal PDGFRA^(Lo) cells support gastric spheroid, but not intestinal organoid, growth in vitro (Fig. 2a, b). To define the niche activities of the sub-glandular fraction, we used flow cytometry to isolate CD55⁺ and CD55⁻ cells from *Pdgfra^(H2BeGFP)* mice (Supplementary Fig. 7b, c), then co-cultured gastric glands with one cell fraction or the other in media lacking recombinant factors. While glands cultured with no mesenchymal cells or with CD55⁻ cells from antral or corpus mesenchyme generated at most a few tiny spheroids, those cultured with CD55⁺ cells robustly stimulated spheroid growth (Fig. 6e). Like unfractionated gastric PDGFRA^(Lo) cells (Fig. 2a), neither fraction induced organoid formation from SI crypts, but both CD81⁺ SI trophocytes and CD55⁺ gastric PDGFRA^(Lo) cells induced spheroid growth from isolated corpus and antral glands (Fig. 6f). To define the signaling pathways most pertinent in the cellular effect of CD55⁺ cells, we added the Porcupine inhibitor Wnt-C59, which impairs secretion of all Wnts[59], or a cocktail of recombinant BMPs 2, 4 and 7, to gastric gland co-cultures. While Wnt-C59 attenuated CD55⁺ cell effects on corpus and antral glands, BMP2/5/7 inhibition was significant on corpus, but statistically insignificant on antral glands (Fig. 6g). Thus, only the CD55⁺ fraction of PDGFRA^(Lo) cells can elicit gastric spheroids, owing in part to functions that BMPs and impaired Wnt secretion can override.

Spheroid growth from gastric glands may reflect an experimental feature (close apposition of mesenchymal cells to epithelium) that differs from the native tissue structure, at least in the corpus where CD55⁺ PDGFRA^(Lo) cells localize far from the isthmus. In the absence of a means to test the function of the full CD55⁺ cell fraction in vivo, *Grem1^(DTR)* mice[35] provided a path to deplete at least the minority *Grem1*⁺ subpopulation. Ablation of intestinal *Grem1*⁺ cells, including PDGFRA^(Lo) CD81⁺ trophocytes, rapidly depletes overlying ISCs and progenitors[33]; in line with this finding, treatment of *Grem1^(DTR)* mice with Diphtheria toxin resulted in >90% and ~50% depletion of *Grem1* and *Rspo3* transcripts (Fig. 7a) and in arrested SI epithelial replication within attenuated crypts (Supplementary Fig. 8b). In contrast, KI67 immunostaining revealed no compromise of corpus or antral epithelial proliferation (Fig. 7b); the only consistent defect was near absence of antral UEAI staining (Fig. 7b), signifying absent or defective basal mucous cells (see also Supplementary Fig. 1b). Ablation of gastric *Grem1*⁺ cells elicited minor reductions in antral RNA levels of *Lgr5* and the proliferative isthmus cell marker *Stmn1*[14], altered *Muc6* levels in corpus and antrum, and reduced *Troy* mRNA in the corpus (Fig. 7b). These findings identify selective and modest effects of *Grem1*⁺ cells on corpus and antral basal cell markers, implying that most of the CD55⁺ cell effect on gastric spheroids reflects *Grem1*⁻ (*Ctgf*⁺) cell activity.

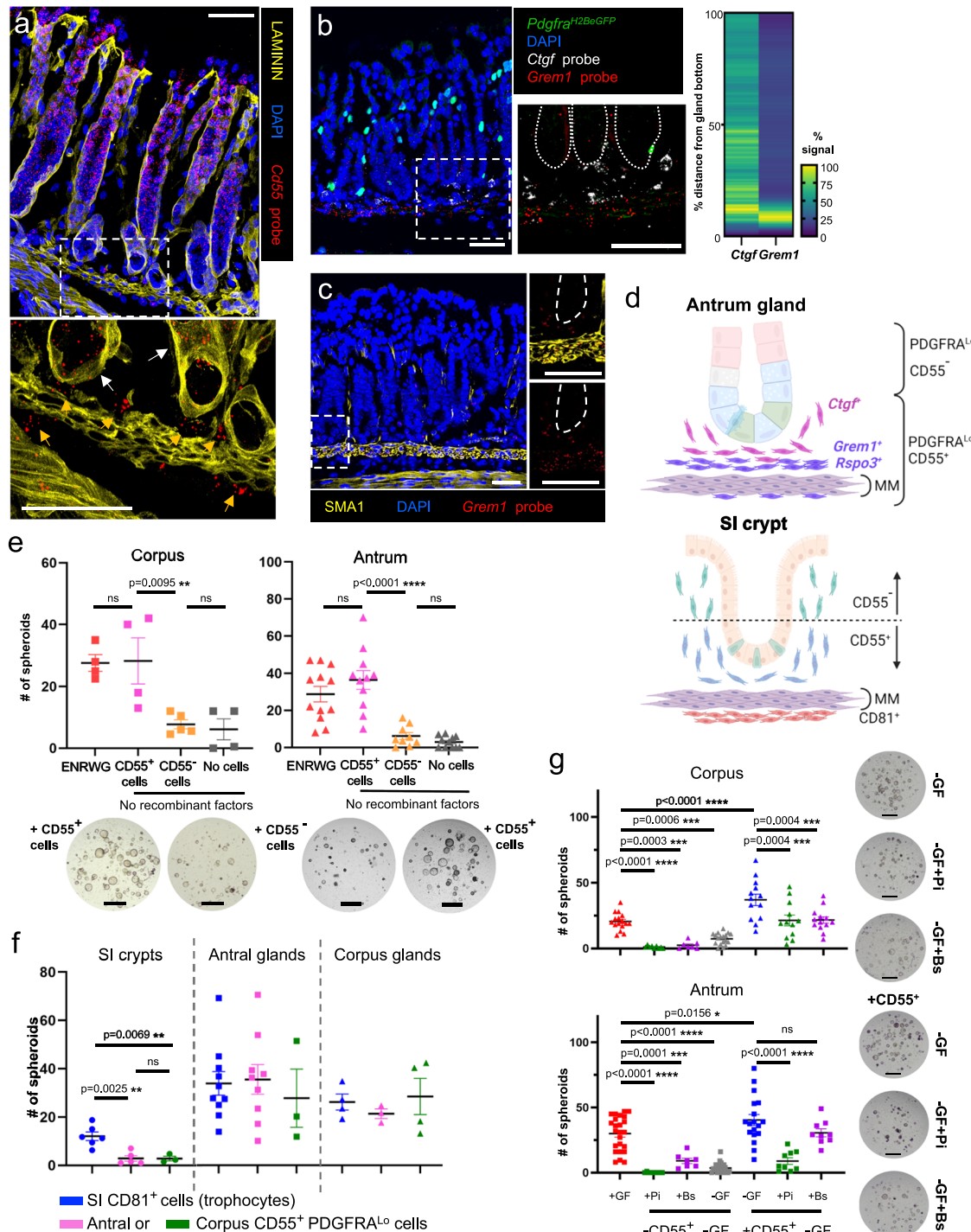

**Fig. 6 | Niche activity of gastric CD55+ PDGFRA^Lo cells.** Antral in situ hybridization localized (**a**) *Cd55* to the epithelium (white arrows) and PDGFRA^Lo mesenchymal cells (orange arrows) near gland bottoms; (**b**) *Grem1* and *Ctgf* near the gland base (dotted lines demarcate glands, heatmaps: fluorescence along individual glands, *n* = 35 to *n* = 39, Source data provided as a Source Data file); and (**c**) *Grem1* in SMA1+ muscularis mucosae, extending into sub-glandular PDGFRA^Lo cells (dashed lines demarcate glands). Dashed boxes are magnified below (**a**) or to the right (**b**, **c**). Images represent fields examined in 2 (**a**, **c**) or 3 (**b**) independent experiments. Scale bars 50 μm. **d** Schema of mesenchymal cell distributions. MM muscularis mucosae. **e** Co-culture of gastric glands with CD55+ or CD55− PDGFRA^Lo cells from each region. ENRWG corpus *n* = 4, antral *n* = 12; CD55+ corpus *n* = 4, antral *n* = 11; CD55− corpus *n* = 5, antral *n* = 9; No cells corpus *n* = 4, antral *n* = 11. Bars: mean ± SEM. Significance: one-way ANOVA and Tukey's multiple comparison test. ns not significant. Source data provided as a Source Data file. Scale bars 400 μm. **f** Spheroid formation from

SI crypts and antral or corpus glands co-cultured with CD55+ (gastric) or CD81+ (SI trophocyte) fractions from each region. SI crypts: CD81+ *n* = 6, Antral CD55+ *n* = 5, Corpus CD55+ *n* = 3; Antral glands: SI CD81+ *n* = 12, Antral CD55+ *n* = 11, Corpus CD55+ *n* = 3; Corpus glands: SI CD81+ *n* = 4, Antral CD55+ *n* = 3, Corpus CD55+ *n* = 4. *n*: independent experiments, each with ≥2 technical replicates. Bars: mean ± SEM. Significance: one-way ANOVA and Tukey's multiple comparison test. ns not significant. Source data provided as a Source Data file. **g** Co-culture of gastric glands with or without corresponding CD55+ PDGFRA^Lo cells, growth factors (+GF or −GF), and Porcupine inhibitor Wnt-C59 (Pi) or BMP2/4/7 cocktail (Bs). Bars: mean ± SEM. No cells+GF corpus *n* = 16, antral *n* = 24; −CD55++Pi corpus and antral *n* = 9, −CD55++Bs corpus *n* = 8, antral *n* = 7; −GF corpus *n* = 18, antral *n* = 25; +CD55+ corpus *n* = 13, antral *n* = 18; +CD55++Pi corpus *n* = 12, antral *n* = 8; +CD55++Bs corpus *n* = 13, antral *n* = 9. Significance: one-way ANOVA and Sidak's multiple comparison test. ns not significant. Source data provided as a Source Data file. Scale bars 400 μm.

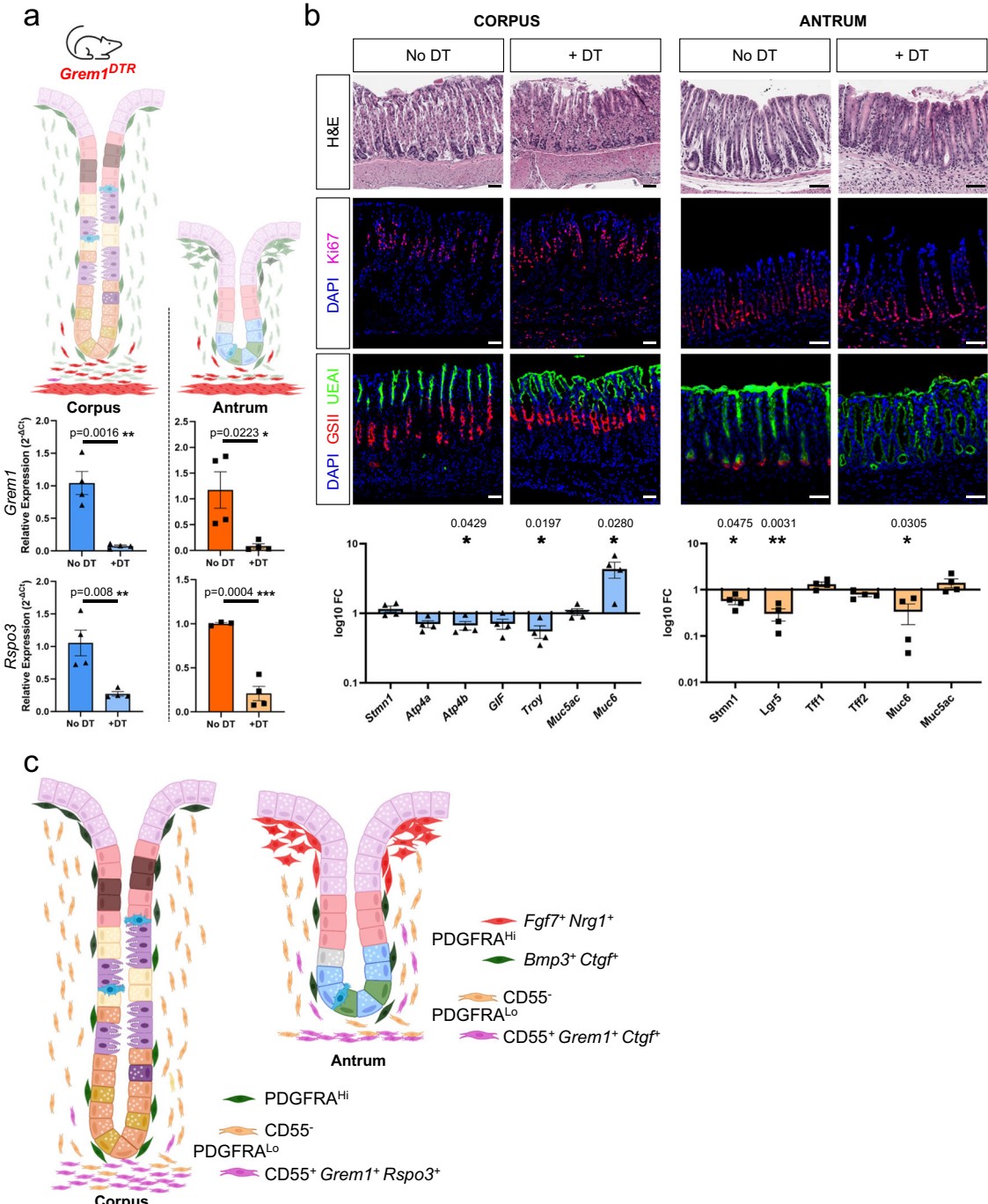

**Fig. 7 | Effects on gastric glands from ablation of *Grem1*-expressing cells.**
**a** *Grem1^{DTR}* mice treated with Diphtheria toxin (DT) and untreated control mice
were used to assess the effects of partial (*Grem1*+ fraction) ablation of the CD55+
PDGFRA^{Lo} cell compartment. *Grem1* and *Rspo3* mRNA levels were significantly
reduced 72 h after DT treatment ($n = 4$). Bars represent mean ± SEM values. Sig-
nificance of differences was determined by two-tailed unpaired Student's *t* test.
**p < 0.05; **p < 0.01; ****p < 0.0001.* **b** Top: histology (H&E), KI67 immuno-
fluorescence, and staining with fluorescent GSII and UEAI lectins to highlight

proliferative and distinct mucous cell types in the corpus and antrum of DT-treated
mice and untreated (No DT) controls. Scale bars 50 μm. Blue, DAPI; magenta, KI67;
red, GSII; green, UEAI. Bottom: qRT-PCR analysis of corpus and antral epithelial cell
markers (relative to *Gapdh* and to untreated controls, no DT) in tissues harvested
72 h after treatment ($n = 4$). Bars represent mean ± SEM values. Significance of dif-
ferences determined by two-tailed unpaired Student's *t* test. Source data are pro-
vided as a Source Data file. **c** Schematic representation of all antral PDGFRA+ cell
types in relation to glandular epithelium.

## Discussion

To ensure a proper epithelial census, microenvironments around
specific zones in intestinal crypts and gastric glands generate Wnt,
BMP, and other signaling gradients that balance cell self-renewal
against differentiation[7,22,28,56,60]. Among mesenchymal cells, PDGFRA+
cells as a group show the most regional diversity. In the intestine, they

provide canonical Wnt ligands[35] and create the requisite BMP
gradient[33], but signaling sources and cellular arrangements are not
known in the gastric corpus and antrum, where gland structures differ
markedly[37]. Our bulk and single-cell RNA profiles, high-resolution
microscopy, and gastric gland co-cultures identify spatially, molecu-
larly, and functionally distinct PDGFRA+ cell types. One large PDGFRA+

cell type in the corpus and antrum expresses high *Pdgfra* and BMP and EGF ligand mRNAs, lies within basement membranes, and fails to induce spheroids in co-culture assays. In these respects, gastric *Pdgfra*[Hi] cells resemble intestinal SEMFs[33,61–63], also called "telocytes"[31,64]. *Pdgfra*[Hi] cell aggregates surround antral gland pits, far from stem cells in gland bottoms and reminiscent of ISEMF density at crypt and villus tops[33,64]. Tight clustering of corpus *Pdgfra*[Hi] cells in nearest-neighbor space suggests that they are likely uniform along the gland length. In contrast, peri-foveolar antral SEMFs differ from corpus SEMFs and deep antral SEMFs in high expression of *Fgf7*, *Nrg1*, and other factors, but *Fgf7* deletion alone had subtle effects on pit cell gene expression. Nevertheless, high and specific *Fgf7* expression in peri-foveolar antral SEMFs and rFGF7 stimulation of budding in vitro suggest possible roles in non-homeostatic conditions.

All other gastric PDGFRA[+] mesenchymal cells express lower levels of *Pdgfra* and BMP and EGF ligand mRNAs, lie farther from the epithelium than SEMFs, and elicit spheroid growth from gastric glands in the absence of supplemental growth factors. In both corpus and antrum, this in vitro trophic activity maps to CD55[+] cells that reside beneath the gland base in vivo. Antral *Cd55* expression and, by extension, in vitro trophic activity is limited to AntLo cells, distinct from *Cd55*[–] AntInt. Furthermore, *Cd55* expression is highly correlated with *Ctgf*, which localizes in the immediate vicinity of antral stem cells. These findings suggest a model wherein antral *Pdgfra*[Lo] cells segregate into a niche-active CD55[+] fraction located near epithelial stem cells and inactive CD55[–] AntInt. The minimal effects from ablation of *Grem1*[DTR] cells in vivo imply that the stem cell support evident in vitro arises from the *Grem1*[–] (*Ctgf*[+]) subpopulation of sub-glandular CD55[+] cells. Nevertheless, anatomic and functional parallels with the sub-cryptal SI stem-cell niche are striking. Furthermore, the gene signature of corpus and antral CD55[+] cells overlaps with that of colonic Fibroblasts 3a ("interstitial fibroblasts") recently defined as a population lying below colonic crypts and expressing *Cd55* and *Grem1* among other genes[65]. The same gene signature also overlaps materially with the resting fibroblast population "F01" recently identified in human gastric cancer specimens[66].

Intestinal trophocytes are CD34[+] CD81[+] PDGFRA[Lo] cells that lie beneath the muscularis mucosae and robustly support spheroid formation from SI crypts;[33,36] however, supportive niche activity extends above the muscularis mucosae, into peri-cryptal CD81[–] CD55[+] PDGFRA[Lo] cells[35]. Thus, common features of antral and intestinal cells with demonstrable in vitro niche activity include PDGFRA (lower than in SEMFs) and CD55 expression, low levels of BMP ligand genes *Bmp2/5/7*, and residence near stem cells. CD55 is a robust surface marker of these uniquely functional cells, in part supplanting CD81, which marks the most potent cells but is useful for cell isolation only from *Pdgfra*[H2BeGFP] mice (PDGFRA Ab only stains SEMFs reliably; most intestinal PDGFRA[Lo] cells escape Ab detection). We therefore propose to redefine "trophocytes" as the niche-active CD55[+] mesenchymal cell fraction in both intestine and antrum. This definition has the practical advantage that CD55[+] mesenchymal cells can be isolated by FACS from any mouse strain, not only from *Pdgfra*[H2BeGFP], after straightforward exclusion of CD45[+] lymphocytes and, for certainty, also EPCAM[+] epithelial cells (*Cd55* RNA is expressed in gastric epithelium, but flow cytometry did not detect the protein on cell surfaces). Notably, both corpus and antral CD55[+] cells triggered spheroid formation from homologous or heterologous glands but not from SI crypts; this may be because *Rspo3* and *Grem1*, essential trophic factors for ISCs, are expressed at lower levels than in intestinal trophocytes. The functions of CD55[–] PDGFRA[Lo] cell fractions, including AntInt and CorpLo3, remain unknown in any region of the digestive tract. CD55 expression is modulated by cytokines in other tissues[67–69]; we investigated CD55 only as a molecular marker and did not ask whether its RNA and protein levels are similarly modulated in gastric epithelium, mesenchyme or lymphocytes.

Constructing a parallel model for corpus *Pdgfra*[Lo] cells poses challenges, in part because a conceptual framework for the mesenchyme's influence on active isthmus and facultative basal stem cells is still lacking. *Pdgfra*[Lo] cells in the corpus are more heterogeneous than those in the antrum, with CorpLo1 clustering far apart from CorpLo2 and CorpLo3 (Fig. 5a), and *Cd55* is expressed in both CorpLo1 and CorpLo2. As in the antrum, *Cd55*-expressing *Pdgfra*[Lo] cells lie between the corpus gland base and muscularis mucosae, but unlike the antrum, this location is not near active isthmus stem cells. Therefore, the effect of corpus CD55[+] cells in spheroid assays might represent their action on facultative stem cells or suggest that they influence isthmus stem cells from a distance in vivo; alternatively, it may reflect a fortuitous consequence of artificial epithelial-mesenchymal cell juxtaposition in spheroid assays. Recalling that *Pdgfra*[Lo] cells along the length of corpus glands lack *Cd55* and are sparse near the isthmus (see Supplementary Fig. 1g), another intriguing possibility is that isthmus stem cells derive their state from a combination of CD55[+] cells acting from a distance and the paucity of CD55[–] cells in their immediate vicinity.

In summary, we report spatial, molecular, and functional features of gastric PDGFRA[+] mesenchyme at high resolution, noting salient differences between gastrointestinal regions (Fig. 7c). Our findings identify CD55 as a robust surface marker of spatially segregated PDGFRA[Lo] cells with demonstrable niche activity in vitro, distinct from CD55[–] cells that are inert in spheroid assays, and establish a foundation for further investigation of gastric mesenchymal niches in homeostasis and disease.

## Methods

### Mouse models and treatment

All animal procedures were approved by Animal Care and Use Committees at the Dana-Farber Cancer Institute or Columbia University Medical Center. Mice were housed in a pathogen-free animal facility and maintained on a 12-h light/dark cycle at constant temperature and humidity, with ad libitum access to food and water. Strains were maintained on a mixed *C57BL/6* background. Male and females aged 8–16 weeks were used. Genotypes were identified by PCR analysis of genomic DNA isolated. *Pdgfra*[H2BeGFP] (JAX strain 007669), *Pdgfra*[Cre(ER-T2)] (JAX strain 032770), *Rosa26*[Lsl-TdTomato] (JAX strain 007908), and *Rosa26*[mT/mG] (JAX strain 07676) mouse strains were purchased from Jackson Laboratories. *Grem1-P2A-DTR-P2A-TdTomato* (*Grem1*[DTR]) mice were described previously[35]. *Rosa26*[mT/mG] mice were crossed with *Pdgfra*[Cre(ER-T2)] to generate compound heterozygotes. Cre recombinase activity was induced by intra-peritoneal (i.p.) injection of 2 mg tamoxifen (Sigma-Aldrich T5648) on 4 consecutive days, followed by visualization of *Pdgfra*-expressing cells. *Fgf7*[fl/fl] mice, generated as described below, were crossed with *Pdgfra*[Cre(ER-T2)] and *Rosa26*[Lsl-TdTomato] to obtain *Pdgfra*[Cre(ER-T2)];*Fgf7*[fl/fl];*Rosa26*[LSL-TdTomato] mice, and tamoxifen 2 mg/day was injected i.p. to induce cell-specific Cre activation and genetic recombination. *Cre*-negative littermates were used as controls. *Grem1*[DTR] mice were treated with two i.p. injections of 50 μg/kg of Diphtheria toxin (DT, Sigma-Aldrich, D0564) on alternate days and harvested 24 h after the second injection.

### Fgf7 conditional mice

A floxed conditional *Fgf7* allele was generated by genome editing in 1-cell embryos. The targeting construct consisted of the *Fgf7* exon 2 sequence flanked by LoxP sites and surrounded by short homology arms in an adeno-associated virus (AAV) vector. This vector was electroporated along with Cas9 protein and two sgRNA, as previously described[70]. Single-nucleotide substitutions at the PAM motifs were included in the targeting vector to prevent re-cutting after homologous recombination. Oligonucleotides used for in vitro transcription of sgRNAs and for genotyping are reported in Supplementary Table 1.

## Gastrointestinal organoid/spheroid culture

Organoids and spheroids were generated from wild-type mouse SI crypts or gastric glands. Crypts were isolated from the first half of the SI by incubating the tissue at 4 °C in cold 2 mM EDTA in phosphate-buffered saline (PBS), as described previously[42]. Crypts were plated in 5 µl Matrigel (Corning 356234) droplets and cultured in ENR medium: DMEM/F12 medium supplemented with 1X Glutamax, 10 mM HEPES, 1X penicillin/streptomycin, 1X normocin (Invivogen, ANT-NR-2), 1X primocin (Invivogen, ANT-PM-2), 1X N2 and B27 supplements (Life Technologies, 17502001 and 17504001), 1 mM N-acetylcysteine, 50 ng/ml murine rEGF (Peprotech, 315-09), 100 ng/ml murine rNOGGIN (Peprotech, 120-10C), and 10% Rspo1 conditioned medium (from 293T cells expressing mouse RSPO1). Glands were isolated from the gastric antrum or corpus by incubating the tissue in 10 mM EDTA in PBS at room temperature, as described previously[71,72]. Glands were separated from the stroma and muscularis by gentle scraping, washed in PBS, and cultured in ENR medium with addition of 10% Afamin/Wnt3a conditioned medium (MBL International, J2-001) and 10 nM gastrin (Sigma-Aldrich, G9145) or as specified in different experiments. Recombinant factors (human FGF10, Peprotech, 100-26; murine FGF7, Peprotech, 450-60; human BMP3, Peprotech, 120-24B; rat CTGF, R&D Systems, 92-37C-T050, Bmp2, Peprotech, 120-02C; murine BMP4, Peprotech, 315-27; human Bmp7, 120-03P) or Porcupine inhibitor Wnt-C59 (Thermo Fisher, NC0186427) were resuspended in 0.1% BSA in PBS and used at final concentrations of 100 ng/ml.

## Quantitation of mRNAs in gastric spheroids and in *Grem1-DTR* mouse tissues

After 4 days of culture, Matrigel droplets were washed in cold PBS, detached gently from the plate, and incubated 20 min in cell recovery solution (Corning, 354270) to depolymerize the matrix. Spheroids were collected by centrifugation, resuspended in Trizol (Thermo Fisher Scientific), and total RNA was isolated with RNeasy Mini Kit (QIAGEN). One µg RNA was reverse transcribed using SuperScript III First-Strand Synthesis System (Invitrogen, 18080-051). Transcript levels were measured using Power SYBR Green (Life Technologies, 4367659) on a LightCycler 480 instrument (Roche) and normalized to *Gapdh* levels ($2^{-\Delta Ct}$). Primer sequences are listed below. Similarly, corpus and antrum from *Grem1*$^{DTR}$ mice treated with Diphtheria toxin and from untreated control animals were collected and processed for RNA isolation, cDNA synthesis, and quantitation of gene expression using primers reported in Supplementary Table 2.

## Mesenchymal cell isolation

SI, antral, and corpus mesenchyme was isolated as reported previously[33,38,54]. Tissue was harvested after perfusing adult *Pdgfra*$^{H2BeGFP}$ mice with cold PBS. After manual removal of external muscles, the epithelium was removed by rotating the tissue at 250 rpm in pre-warmed chelation buffer (10 mM EDTA, 5% fetal bovine serum (FBS), 1 mM Dithiothreitol, 10 mM HEPES in Hank's Balanced Salt Solution, HBSS) for 20 min at 37 °C. The tissue was then washed with PBS and digested for 10 min at 37 °C in 3 mg/ml collagenase II (Worthington, LS004176) in HBSS supplemented with 5% FBS and 10 mM HEPES. Cells were harvested by centrifugation at 300 × g for 5 min For bulk RNA sequencing, cells were sorted on a BD FACSAria II cell sorter, gating on DAPI negative cells to identify live cells, and GFP to sort *Pdgfra*-high or -low expressing cells.

## Epithelial cell isolation

To isolate epithelial cells, corpus and antrum were harvested from adult *Pdgfra*$^{H2BeGFP}$ mice perfused with cold PBS. Glands isolated as described above were digested first for 10 min at 37 °C in 3 mg/ml collagenase II (Worthington, LS004176) and 4X TrypLE (Thermo Fisher, A1217701) in DMEM supplemented with 1X penicillin/streptomycin on a rotating shaker, followed by incubation in 0.1 mg/ml DNase

I (Worthington, LS002139) at 37 °C for an additional 5 min Enzymes were inactivated in DMEM + 10% FBS. Cells were collected by centrifugation (500 × g for 5 min), resuspended, and stained for FACS analysis. For scRNA-seq, cells were sorted on a BD FACSAria II cell sorter, gating on DAPI negative cells to identify live cells. For CD55 expression experiment, cells were stained with APC-conjugated EpCAM antibody (BioLegend, 118214, 1:100) and PE-conjugated CD55 antibody (BioLegend, 131803, 1:75).

## Gland and crypt co-cultures with mesenchymal cells

Whole mesenchyme was isolated and plated on tissue culture plates (SI) or fibronectin-coated tissue culture plates (gastric mesenchyme, Corning, 354451) in DMEM/F12 medium supplemented with 10% FBS, 1X penicillin/streptomycin, 1X normocin, 1X primocin, 1X Glutamax, and 10 mM HEPES. Three days later, cells were washed in PBS, removed from the plates with 0.25% Trypsin (5 min at 37 °C), washed, and stained with antibodies. SI mesenchyme was stained with biotin-CD81 antibody (eBioscience, 13081181, 1:100) followed by incubation with streptavidin-APC-conjugated secondary antibody (eBioscience, 17431782, 1:100). Gastric cells were incubated with PE-conjugated CD55 antibody (BioLegend, 131803, 1:75) or, to isolate endothelial cells, with PE-conjugated CD31 antibody (BD Biosciences, 553373) and PE-Cy7-conjugated CD45 antibody (Invitrogen, 25-0451-82). FACS gating strategies are reported in Supplementary Fig. 9a. In total, 5000 viable FACS-sorted cells were co-cultured with ~20 glands or crypts in 5 µl Matrigel droplets in basal media (DMEM/F12 supplemented with 1X Glutamax, 10 mM HEPES, 1X penicillin/streptomycin, 1X normocin, 1X primocin, 1X N2 and B27 supplements, and 1 mM N-acetylcysteine) with no recombinant factors or as specified.

## Immunohistochemistry and FACS analysis

Immunohistochemistry was performed on 10 µm fixed tissue sections prepared using a Leica cryostat on tissues frozen in OCT compound (Tissue-Tek, VWR Scientific, 4583). Whole-mount immunohistochemistry was performed as described[73]. Mice were perfused with ice cold PBS, followed by 4% paraformaldehyde (PFA). Antrum or corpus was pinned on agarose plates, treated with mucolytic solution (20 mM N-acetyl cysteine and 20 mM DTT in PBS) for 15 min at room temperature, and fixed for 15 min in 4% PFA. External muscle was removed manually before overnight fixation with 15% picric acid, 0.5% PFA in PBS. On day 2, tissues were washed in PBS and placed in 10% sucrose for 4 h, then 20% sucrose overnight at 4 °C. On day 3, tissues were placed in blocking buffer (0.5% bovine serum albumin, 0.6% Triton X-100, 0.05% goat serum, 0.0005% sodium azide) for 6 h, followed by overnight incubation with primary antibodies at 4 °C. On day 4, after 5 hourly washes in wash buffer (0.3% Triton X-100 in PBS), tissues were incubated at 4 °C with Alexa Fluor-conjugated secondary antibodies (Invitrogen) and DAPI (1:1000 dilution). The next day tissues were washed for 5 h with buffer changes every 30 min and fixed in 4% PFA at 4 °C for 1–2 days. Hand-cut 1 mm fragments were positioned on glass slides with imaging spacers (Grace Bio-Labs SecureSeal, Sigma-Aldrich), cleared for 30 min with FocusClear (CelExplorer, FC-101) at room temperature, and mounted with VectaShield (Thermo Fisher Scientific) medium. Tissues from at least three independent mice were imaged using an SP5X laser scanning confocal microscope (Leica Microsystems CMS GmbH) or an LSM980 microscope and ZEN Desk 3.4 software (Carl Zeiss GmbH). Images were analyzed using Fiji[74] and videos were generated with ZEN Desk 3.4 software. The following primary antibodies were used at 1:500 dilution unless otherwise specified: CD31 (BD Biosciences, 557355); Laminin (Sigma-Aldrich, L9393); alpha-smooth muscle actin1 (SMA1, Abcam, ab5694); PDGFRA (R&D Systems, AF1062, 1:100); MUC5AC (Cell Signaling Technology, 6119S); Gastrin (Abcam, ab232775, 1:100); and Somatostatin (Santa Cruz Biotechnology, sc-55565, 1:100). Alexa fluor 647-conjugated lectin GS-II (1:500, Life Technologies, L32451) and FITC-conjugated lectin UEAI

(1:500, Life Technologies, L32476) were used to identify gastric mucous cell types.

CD55 expression was analyzed on freshly isolated mesenchymal cells. Whole mesenchyme was isolated as described above. Cells were blocked for 15 min on ice with TruStain FcX PLUS (BioLegend, 156603) and stained for 20 min at room temperature with the following antibodies: BV711 anti-CD11b (BD Biosciences, 563168), Alexa Fluor 647 anti-CD19 (BioLegend, 115525), PE/Cy7 anti-CD3 (BioLegend, 100219), APC/Cy7 anti-CD45 (BioLegend, 103115), PE anti CD55 (BioLegend, 131803). After two washes, cells were incubated with DAPI (BD Biosciences, 564907) and analyzed with a BD LSR Fortessa cell analyzer. The gating strategy is reported in Supplementary Fig. 9.

## RNA sequencing

For bulk RNA sequencing, *Pdgfra*-expressing cells were isolated, FACS-sorted from antrum and corpus as described above, and placed in Trizol Reagent (Thermo Fisher Scientific). Total RNA was isolated with RNeasy Micro Kit (QIAGEN) and used for library preparation (5–10 ng) using SMART-Seq v4 Ultra Low Input RNA Kit (Clontech). For single-cell RNA sequencing, whole mesenchyme was isolated as described above and dead cells were eliminated by flow cytometry with DAPI stain. About 10,000 cells were loaded onto a 10X Genomics Chip G. Libraries were prepared according to the manufacturer's protocol using Single-Cell 3′ v3.1 chemistry (10X Genomics, PN-1000128) and Libraries were sequenced on the Novaseq platform (Illumina).

## RNAscope in situ hybridization

mRNA detection for spatial localization of cell populations was carried out using the RNAscope Multiplex Fluorescent Reagent Kit v2 (Advanced Cell Diagnostics)[33,75]. and probes designed by Advanced Cell Diagnostics: *Cd55* (421251), *Grem1* (314741), *Ctgf* (314541), *Fgf7* (443521), *Bmp3* (428461), *Nrg1* (468841), and *Rspo3* (483781). We followed the manufacturer's recommended procedure, including 3 min (corpus) or 5 min (antrum) of target retrieval. As the procedure extinguished native GFP fluorescence, GFP signals were revealed by immunohistochemistry using GFP antibody (Abcam, ab6556, 1:100) as described previously[33]. Smooth muscle actin (SMA1, Abcam, ab5694) and Laminin (Sigma-Aldrich, L9393) antibodies were also used (1:100). Tissues were imaged using an SP5X laser scanning confocal microscope (Leica Microsystems CMS GmbH) or an LSM980 microscope and ZEN Desk 3.4 software (Carl Zeiss GmbH). Fluorescent signals along gastric glands were quantified with Fiji software.

## Data processing and analysis

**Image segmentation and quantitation.** GFP⁺ cells were quantified using Cellpose plugin in CellProfiler[76,77]. Eight images were quantified per gastric region and representative examples are reported. Masks were generated using Cellpose pre-trained model "cyto2" and a cell diameter of 43 pixels. Each image contained about 300 segmented GFP⁺ cells and >99% of cells were included in the segmentation. Segmented masks and original images were then fed into CellProfiler to quantify the distribution of GFP⁺ cells. Signal intensity cut-offs to distinguish GFP$^{Hi}$ from GFP$^{Lo}$ cells were established from analysis of multiple cells. Data were represented graphically using Prism9 (GraphPad Inc).

**Bulk RNA-sequencing.** Data were aligned to mouse reference genome mm10 using the Viper pipeline with default settings[78] and the quality was verified using RSeQC[79]. The R platform 4.2.1 was used for further analyses. Reads from genes known to express in gastric epithelium and genes encoded on sex chromosomes were excluded. Data were normalized and differential gene expression ($p_{adj} < 0.05$; |log₂ foldchange| > 1.5) was determined with the DESeq2 package[80]. Reads per kb per million tags (RPKM) values were generated from normalized counts. Pearson correlation coefficients were calculated from DESeq2

normalized counts and plotted using Corrplot package[81]. Principal component analysis (PCA) was done and heatmaps of selected genes were generated using Rlog-transformed counts. Integrative Genome Viewer (IGV) data tracks were generated from RPKM-normalized bigwigs loaded into IGV 2.15.4 (Broad Institute). Data from SI mesenchyme (GSE130681) were reported previously[33].

**Single-cell RNA sequencing (scRNA-seq).** Samples were aligned to the mm10 genome using 10x Genomics Cell Ranger 3.0.2 with default parameters[82]. Bioinformatic analyses were conducted using R version 4.2.1. Single-cell analyses used Seurat 4.1.0[83]. Cells were filtered for <12% mitochondrial and >1000 unique reads. Data were merged and normalized using sctransform[84]. Cell features were detected with the FindVariableFeatures function (2000 features, selection method "vst"). The first 15 principal components were taken to run uniform manifold approximation and projection (UMAP) algorithms[85]. Cell types were identified using established marker genes detected by the "FindAllMarkers" function in Seurat2 (min.pct = 0.25, test "roc").

To assess similarities between gastric mesenchymal cell types, we used MetaNeighbor-AUROC analysis. Highly variable genes in the datasets were used to train an algorithm to predict the type of a given cell by the genes expressed in its nearest neighbors. To calculate similarity between cell types A and B across any pair of samples, the cell type annotation from one group was hidden and a score was used to represent how often a cell of type A is predicted to overlap with cells of type B vs. all other cell types (area under the receiver operating characteristic, AUROC score). Higher scores indicate greater similarity between two cell types.

To address concerns about separation of *Pdgfra*⁺ populations, we used AUGUR analysis[49], which uses a regularized random forest classifier to directly quantify transcriptional differences between cell types under different conditions. High AUC scores indicate larger transcriptional difference between a given cell type in condition A vs. condition B (in this case corpus vs. antrum).

Previously published scRNA-seq data from corpus epithelium were analyzed to determine *Cd55* expression[14,21,86]. Cells were filtered for >200 unique reads and mitochondrial content <40%. Data were integrated based on a set of anchors between Seurat objects found using FindIntegrationAnchors, normalized using sctransform and analyzed as explained above.

**Statistics.** Prism9 (GraphPad Inc) was used for statistical analysis of all data. Specific tests and significance levels are indicated in figure legends. Graphical drawings were created at BioRender.com.

No statistical method was used to predetermine sample size. No data were excluded from the analyses. The experiments were not randomized, and the investigators were not blinded to allocation during experiments and outcome assessment.

Imaging experiments were repeated a minimum of three times with comparable results and representative images are reported in the study.

## Reporting summary

Further information on research design is available in the Nature Portfolio Reporting Summary linked to this article.

# Data availability

The sequencing data generated in this study have been deposited in the Gene Expression Omnibus (GEO) repository under accession code GSE224737. This study includes analysis of previously published data, referenced with GEO accession number GSE130681, ArrayExpress accession numbers E-MTAB-6879 (https://www.ebi.ac.uk/biostudies/arrayexpress/studies/E-MTAB-6879?query=E-MTAB-6879%20), and NCBI Sequence Read Archive SRP227356 (https://www.ncbi.nlm.nih.gov/sra/SRX7069341%5baccn%5d). Any additional information

required to reanalyze data reported in this paper is available from the corresponding author upon request. Source data are provided with this paper.

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

## Acknowledgements

This study was supported by NIH grant RO1DK121540 (R.A.S.), the Intestinal Stem Cell Consortium of NIDDK and NIAID (U01DK103152 to R.A.S. and U01DF103155 to T.C.W.), NIH fellowship K01DK125639 (N.M.), and a gift from the Sarah Rhodes Fund (R.A.S.). We thank Ke Li for analyzing published corpus epithelial scRNA-seq data and gratefully acknowledge the contributions of molecular imaging, flow cytometry, and organoid core facilities in the Harvard Digestive Diseases Center and the Dana-Farber Cancer Institute.

## Author contributions

E. Manieri, N.M. and R.A.S. conceived and designed the study; E. Manieri, G.T., E. Malagola, A.M. and N.M. performed experiments and interpreted data; E. Manieri, S.M. and R.H. performed computational analyses; D.S., K.Z., Y.F. and S.H.O. generated the Fgf7fl/fl mouse model; K.H. assisted with image signal quantitation; T.C.W. and R.A.S. supervised aspects of the study; E. Manieri and R.A.S. wrote the manuscript, with input and approval from all authors.

## Competing interests

The authors declare no competing interests.
