## [Peer Review File · Nature Communications]

Role of PDGFRA+ cells and a CD55+ PDGFRA^{Lo} fraction in the gastric mesenchymal nicheREVIEWER COMMENTS

Reviewer #1 (Remarks to the Author):

Manieri et al. use high-resolution imaging and sequencing approaches (bulk RNA and scRNA) to analyze subpopulation and niche properties of mouse corpus and antral mesenchyme, specifically PDGFRA+ cells. By using these complementary approaches, the authors could gain unprecedented insight into the molecular and functional organization of the gastric mesenchyme. Taken together, the study is of high value for the gastric scientific community.

Major comments:

- The authors failed to localize the Grem1 and Ctgf populations in the corpus. It is not readily conclusive that this is “likely owing to high mucus content”, as the antrum also contains mucus producing cells and the probes are targeting non-epithelial cells. Did the authors try other in situ markers of the populations? Or antibodies? Or spatial RNASeq? Taken together the authors should aim to develop a similar spatial model of PDGFRA positive cells for the corpus (and sketch it as for the antrum in Fig 5h), and not merely state that “the corpus has a similar functional counterpart”. This model should trigger in vitro assay that verify potential differential influence of these corpus PDGFRA populations on the epithelial cells in proximity, especially the proliferative isthmus stem cells and the quiescent Lgr5/Troy positive reserve stem cell population at the gland base (Lee JH et al, PMID: 35523142).

Minor comments:

- Figure 1E: The authors should also present stomach tissues before tamoxifen administration as a control (and exclude Tam induced toxicity).
- The medium experiments using the organoids in Figure 2A and Extended Figure 2A, B are not really necessary for the overall story, this data might be removed to make the story more straight forward, or the authors should elaborate more on the relevance.
- How was the area of organoids analyzed? The authors should explain this in the Material and Methods part.
- Page 5, lines 146-148: Please explain better why it is expected that Hox genes are enriched in SEMFs and PDGFRA-Low cells?
- Text and Figure 3C: Please introduce the role of CD34 in PDGFRA+ cells.
- The authors show many different IF stainings and use a large number of markers. It should be explained better in the manuscript which marker stains which kind of cell type. For example, CD31 staining for blood vessels: The manuscript text only mentions that blood vessels have been stained (page 3, lines 71-73). Here, it would be easier for the (non-mesenchyme expert) reader if this would be mentioned in the main text. Also, an explanation of CD34, CD81, SST, and GAST would be very helpful for the reader.
- Extended Figure 1G: Did the authors add WNT to that assay?
- Wrong Figure citation on page 4 line 11: Fig 2a should be Extended Figure 2a
- Figure 2C: It would be good to present the data of the corpus versus the SI also in the main figure and not only in the Extended Data Figure. Maybe another part of the figure can be moved to the supplements instead? Or c) made a little bit smaller?
- Some figure and captions are of poor quality and hard to read. The quality of the final figures should be improved.
- Suppl. Fig 1e: Does the quantification of distribution of PDGFRA cells in the antrum and corpus contain both hi and low cells? Please clarify. Both populations need to be analyzed separately.
- What is the additional information that can be derived from the PDGFRA-

CreERT2/Rosa26-mTmG experiment? Please elaborate on this.

- Fig 1g and Suppl Fig 1g: In my view, the data of Suppl. 1g is more interesting to the story and should be flipped with the data of Fig 1g. What happens if Wnt is left out of the medium in the PDGFRA low co-cultures? The wording “from different gastric segments” is a bit unclear, why not write “from antrum and corpus”?

Reviewer #2 (Remarks to the Author):

Using high-resolution imaging and sequencing, Manieri and colleagues characterized subpopulations and niche properties of mouse corpus and antral PDGFRA+ cells. They report that PDGFRA^{Lo} CD55+ cells enable corpus and antral organoid growth in contrast to CD55- cells. The study shows a comprehensive characterization of CD55+PDGFRA^{low} cells. While this study is mostly descriptive, a little more depth would be required to strengthen the claims and better understand the function of these cells within the stem cell niche. Major points are described below.

- Grem1 expression is quite low in PDGFRA^{low} cells in Antrum and Corpus, and other cells have been described to have prostemness niche function, for example lymphatics. Are CD55+ cells the only population to have niche function in their model, including other cells close to stem cells such as PDGFRA⁻ and CD31+ populations?

- Addition of stromal cells to intestinal organoids has been shown to induce spheroids. Analysis and quantifications in vitro should discriminate between organoids and spheroids.

- On which factor depend the effect of CD55+ cells on organoids/spheroids? Is this effect dependent on Grem1?

- To clarify the function of this population, it would be useful to provide additional evidence that these cells are essential and required to maintain stem cells in vivo. This could be done by performing depletion studies for example with LepR reporter mice, as LepR seem to be expressed in these cells.

- A population of PDGFRA^{low}CD55+ that express Grem1 has already been reported (Jasso et al., 2022), although in the colon. How similar are these populations in terms of gene signature?

- CD55 expression has been described on several other cell types. The claim that CD55 alone is sufficient to identify niche stromal cells should therefore be further investigated. Detailed FACS analysis for CD55 in different subsets of CD45+ (including myeloid and lymphoid gates) and CD45- cells (including PDGFRA⁺ and PDGFRA⁻ cells) and epithelial cells from the lamina propria could help strengthen this claim.

- CD55 is a surface marker, which can be upregulated and downregulated on niche cells. What regulate CD55 expression on these cells? Any evidence that expression of CD55 is faithful to this population at different ages and conditions (notably inflammation)?

- What is the lineage relationship with the other subsets of PDGFRA⁺ and other progenitor cells that have been described in the lamina propria? Pseudotime analysis could help answer this question.

- More information on the crosstalk of this population with surrounding cells would help increase our knowledge of the niche, notably with immune and vascular cells.

- Is this population found in the human gastric mesenchymal niche, at steady-state and in chronic pathologies such as gastric cancer?

Reviewer #3 (Remarks to the Author):

Manieri et al., 2023 characterized the mesenchymal niche of the murine gastric epithelium which has not been thoroughly studied so far. The authors divided the corpus and antrum regions and compared the mesenchymal profiling of the corpus, antrum, and small intestine to comprehensively characterize each mesenchymal niche. Similar to their previous study (McCarthy et al., 2020), imaging, bulk RNA sequencing of PDGFRA⁺ cells, single-cell RNA sequencing, in situ hybridization, and organoid experiments were used. They discovered that the PDGFRA⁺ cells, particularly the PDGFRA [low] cells, are sufficient to support epithelial stem cell growth in both corpus and antrum, while the PDGFRA [high] SEMF cells express potential niche signaling elements. They finally found out that Cd55, a membrane protein, is expressed in the PDGFRA[low]-specific manner, and postulated that it could be a useful molecular marker for isolating PDGFRA[low] cells in both the corpus and antrum. Even though the data, including the scRNA-seq, is informative and has great potential to be a useful resource in the field, the current manuscript has a significant number of concerns and critical points that need to be addressed.

1. The majority of references regarding stem cell populations in the stomach are absent. Especially, this manuscript only cites Karam, S.M.'s 1993 papers for the introduction of the corpus stem cells. Consequently, the current description of stem cells is wrong and the interpretation of their data is too simplistic. Many parts must be revised or rewritten in light of the following papers. As the manuscript defines the niches for stem cells and differentiated cells, the precise introduction of the most up-to-date information regarding stomach stem cells is required.

In the corpus epithelium, two distinct stem cell populations have been identified: fast-cycling isthmus stem cells and the most-quiescent chief cells (PMID: 31422913, PMID: 31589873). The isthmal stem cells control the pit, isthmus, and upper neck regions, while the chief cells control the lower neck and chief cell regions in homeostasis (PMID: 31422913, PMID: 31589873). In homeostasis, the chief cells express Wnt pathway genes, such as Troy (PMID: 24120136) and Lgr5 (PMID: 28581476); upon injury, the chief cells become activated to replenish the cell loss (PMID: 20854822, PMID: 24120136, PMID: 28581476, PMID: 32768422, PMID: 29467218, PMID: 34497145, PMID: 35523142).

Given this information, the following parts in the introduction should be particularly revised: In lines 41-43, 'whereas the gastric corpus epithelium turns over slowly, ~in the isthmus of the each glandular unit' is the wrong description. Line 45 'Antral stem cells express Lgr5 and concentrate near the gland base' misses the citation (Barker et al., 2010; PMID: 32025032).

A recent paper about antral stem cells is also missing (PMID: 32025032).

I believe the authors could analyze their data and extract more intriguing observations based on the background information from the missing references. The current version does not investigate the corpus mesenchymal niche in depth due to the absence of these references.

2. The title 'Defining the structure, signals, and cellular elements of the gastric mesenchymal niche' could mislead the readers. As the manuscript is mainly focusing on the PDGFRA+ niche, the addition of the PDGFRA+ niche in the title should be better.

3. In Fig 1g, they argued that 'SI and gastric stem cells have distinct cellular support requirements', but their experiments are not enough to support this argument. 1) They did not assess if antral or corpus glands can form the organoids when supported by SI PDGFRA [low] cells. 2) There is a possibility that other niche stromal cells (or a mixture of other niche stromal cells with PDGFRA [low] cells from heterologous origin) can support the heterologous epithelial stem cells but it was not addressed as well. Unless these experiments are performed, it is hard to tell if the cellular support requirements are distinct.

4. They used the same media cocktail (ENRW) for SI organoids and stomach organoids. However, it is well known that the stomach organoids need two more components, Gastrin and Fgf10 for their maintenance (PMID: 33223522). This is also shown in the reference they cited (Barker et al., 2010). They should change the media condition to the right one.

5. They did not properly describe what the 'organoid number' and 'organoid area' mean (Fig 2, Extended Data Fig. 2) and how the quantification was performed. How did they match the starting number of cells? Did they isolate the single cells and seed the same number of cells in each dome? Are organoid area and organoid size the same? If the organoid area means the total area taken by all the organoids, the organoid area can be affected by the number of organoids. Is this the best measurement tool to show the difference in growth and forming efficiency of the organoids?

6. In Fig. 3, scRNA seq was performed on all stromal cells in the corpus and antrum; however, only pdgfra+ populations were analyzed further because they displayed a lower degree of similarity in pdgfra+ populations between the corpus and antrum (Fig. 3b). Indeed, the UMAP clearly separates the pdgfra+ cells in the corpus and antrum (Fig. 3c). Nonetheless, this type of distinct separation may occur if the batch effect of the experiment is not corrected. If cells are processed separately on distinct chips, the batch effect can occur. The authors should explain in the methods section how they corrected the batch effect and how they came to the conclusion that the separation of pdgfra+ populations between the corpus and antrum is based on their actual characteristics.

7. The result section of Fig. 3c states that the Pdgfra+ cells contain 6 clusters, but the UMAP in Fig. 3c displays only 4 colors (purple, red, blue, and green). What are these colors based on? Authors should include a suitable annotation and proper explanation of this mismatch.

8. Even though the authors discovered multiple DEGs in each cluster using scRNA-seq, they validated just a handful of genes. Specifically, there are no validation data in the corpus region, and the researchers stated that RNA Scope failed ('In situ hybridization was unreliable in the corpus' in line 308). As they still have the option to use IHC, they should exert more effort to provide readers with beneficial information regarding the spatial expression pattern of the potential marker genes.

9. The authors divided the antral SEMF (Pdgfra [high]) into two populations: Fgf7+ Nrg1+

SEMF in the upper glands adjacent to pit cells and Bmp3+ Ctgf+ SEMF in the lower glands adjacent to proliferative cells. Given that the Bmp3+ Ctgf+ expressing-SEMF subpopulation is in close proximity to the proliferative cells, the authors sought to determine whether SEMF can promote antral gland growth (proliferation) in vitro in Fig. 4d. The PDGFRA[high] cells (GFP[High] cells from Pdgfra[H2BeGFP] antrum) were cocultured with the antral glands due to the inability to identify an appropriate surface marker isolating Bmp3+ Ctgf+ SEMF exclusively. Due to two main factors, they cannot reach the conclusion stated on pages 255-256 using this experiment. 1) PDGFRA[high] cells must be Fgf7+ Nrg1+ SEMF and Bmp3+ Ctgf+ SEMF positive. As Fgf7+ Nrg1+ SEMFs are found in close proximity to differentiated pit cells, they may function to promote epithelial cell differentiation. Therefore, it is not possible to rule out the possibility that Fgf7+ Nrg1+ SEMF reduces the effect of Bmp3+ Ctgf+ SEMF. 2) The author demonstrated that PDGFRA[high] cells alone are insufficient to sustain stem cells. If there is no niche factor for stemness, then organoids cannot form in the first place; consequently, there is no organoid growth (proliferation). Therefore, the authors must redesign the experiment to include minimal niche factors (Wnt agonists) and investigate the difference in proliferation according to the presence of Bmp+ Ctgf+ SEMF.

10. What is the connection between the subpopulations in Figure 3c (CorpLo, CorpLo1, AntInt, AntLo) and Figure 5a (CorpLo1, CorpLo2, CorpLo3, AntInt, AntLo)? As these subclusters were extracted using the same scRNA seq data, there may be a correlation between them. Otherwise, they must annotate each cluster with a unique name so that readers can comprehend the data correctly.

<Minor points>

1. Proper annotation including cell types in the corpus and antrum glands in Fig 1a is needed.
2. Laminin staining cannot directly show the gland pit. Pit cell marker staining using Muc5ac antibody or UEAI will be the proper marker for pit cells.
3. Errors in the Figure citation:
 - in line 81, Fig. 1a -> Fig. 1d
 - in line 111, Fig. 2a -> Extended Data Fig. 2a
4. In lines 214-216, the Wnt4 transcript level is not shown in the corresponding figures.
5. The remaining GFP in Fig. 5c cannot correctly reflect the original GFP expression. Authors could use GFP probes to show GFP expression levels.

Response to Reviewers

Reviewer 1

Manieri et al. use high-resolution imaging and sequencing approaches (bulk RNA and scRNA) to analyze subpopulation and niche properties of mouse corpus and antral mesenchyme, specifically PDGFRA⁺ cells. By using these complementary approaches, the authors could gain unprecedented insight into the molecular and functional organization of the gastric mesenchyme. Taken together, the study is of high value for the gastric scientific community.

Response: We thank the Reviewer for noting the strengths of the study and for making specific suggestions to improve the manuscript.

Major comments:

- The authors failed to localize the Grem1 and Ctgf populations in the corpus. It is not readily conclusive that this is “likely owing to high mucus content”, as the antrum also contains mucus producing cells and the probes are targeting non-epithelial cells. Did the authors try other in situ markers of the populations? Or antibodies? Or spatial RNASeq? Taken together the authors should aim to develop a similar spatial model of PDGFRA positive cells for the corpus (and sketch it as for the antrum in Fig 5h), and not merely state that “the corpus has a similar functional counterpart”. This model should trigger in vitro assay that verify potential differential influence of these corpus PDGFRA populations on the epithelial cells in proximity, especially the proliferative isthmus stem cells and the quiescent Lgr5/Troy positive reserve stem cell population at the gland base (Lee JH et al, PMID: 35523142).

Response: This is a fair concern. On the Reviewer’s recommendation, we overcame previous technical limitations with RNA in situ hybridization (ISH) in corpus tissue (immunohistochemistry with multiple CD55 antibodies was not interpretable owing to high background). Using ISH probes for the co-expressed markers Grem1 and Rspo3, we localize corpus CD55⁺ PDGFRA^{Lo} cells to the sub-glandular mesenchyme, i.e., between gland bases and the muscularis mucosae, as we found in the antrum. Despite their distance from the isthmus *in vivo*, these isolated cells induced spheroids when co-cultured with corpus glands, again similar to the antral counterparts and distinct from CD55⁻ PDGFRA^{Lo} cells, which were inert in that assay (Fig. 2b). To better understand the basis of this effect, we treated co-cultures with a Porcupine inhibitor (Wnt-C59) or with a BMP cocktail (BMPs 2, 4 and 7), which attenuated but did not abrogate, corpus spheroid formation (Fig. 6d). Although these findings do not lead to a complete model for the corpus mesenchyme (not readily achieved in a single study), they do improve the manuscript, we discuss the findings critically (p. 13,14), and we valued the Reviewer’s thoughtful critique.

Minor comments:

- Figure 1E: The authors should also present stomach tissues before tamoxifen administration as a control (and exclude Tam induced toxicity).

Response: We appreciate this suggestion and have added representative images showing that the only discernible change in stomach tissue before and after administration of 8 mg (2 mg

doses in 4 sequential days) Tamoxifen is the appearance of GFP expression in PDGFRA⁺ cells (new Supplementary Fig. 2).

- The medium experiments using the organoids in Figure 2A and Extended Figure 2A, B are not really necessary for the overall story, this data might be removed to make the story more straight forward, or the authors should elaborate more on the relevance.
- How was the area of organoids analyzed? The authors should explain this in the Material and Methods part.

Response: We appreciate this suggestion to tighten the story. Organoid experiments with different media (and corresponding data on organoid area) are no longer part of the revised manuscript. We note for the Reviewer's interest that we use CellProfiler to create a mask and measure the total area occupied by organoids in each Matrigel droplet.

- Page 5, lines 146-148: Please explain better why it is expected that Hox genes are enriched in SEMFs and PDGFRA-Low cells?

Response: We had not meant that one expects Hox genes to be enriched in PDGFRA⁺ cells but rather that expression of anterior and posterior Hox genes distinguishes gastric from intestinal cell sources, respectively, and therefore gave us confidence in tissue isolations. We have reworded the sentence for clarity.

- Text and Figure 3C: Please introduce the role of CD34 in PDGFRA⁺ cells.

Response: Thank you for this suggestion. The revised text now states that intestinal PDGFRA^{Lo} cells, but not SEMFs, express CD34 (p. 7). Later in the text, we considered CD34 as a potential marker for niche-active PDGFRA^{Lo} subpopulations and explain why it is inferior to CD55 in this respect (p. 10, Fig. 5c).

- The authors show many different IF stainings and use a large number of markers. It should be explained better in the manuscript which marker stains which kind of cell type. For example, CD31 staining for blood vessels: The manuscript text only mentions that blood vessels have been stained (page 3, lines 71-73). Here, it would be easier for the (non-mesenchyme expert) reader if this would be mentioned in the main text. Also, an explanation of CD34, CD81, SST, and GAST would be very helpful for the reader.

Response: This is an excellent suggestion. New Supplementary Fig. 1a provides a table of all markers we used to identify specific gastric cell populations or structures.

- Extended Figure 1G: Did the authors add WNT to that assay?

Response: We did not add WNT to the assay shown in Fig. 1g (now Fig. 2a), as we now clarify in the Methods and figure legend. Unless specified differently, all co-culture experiments were carried out in basal media, without supplemental recombinant factors.

- Wrong Figure citation on page 4 line 11: Fig 2a should be Extended Figure 2a

Response: Thank you for noting the error, which we have corrected.

- Figure 2C: It would be good to present the data of the corpus versus the SI also in the main figure and not only in the Extended Data Figure. Maybe another part of the figure can be moved to the supplements instead? Or c) made a little bit smaller?

Response: We have adjusted the figure as recommended (Fig. 2d)

- Some figure and captions are of poor quality and hard to read. The quality of the final figures should be improved.

Response: We apologize sincerely. The original figures were of excellent quality but their conversion to PDF during submission was sub-par. We missed that degradation in our review of the final submission and will ensure that it does not repeat.

- Suppl. Fig 1e: Does the quantification of distribution of PDGFRA cells in the antrum and corpus contain both hi and low cells? Please clarify. Both populations need to be analyzed separately.

Response: This is an important point. In the revised manuscript, we quantify PDGFRA^{Hi} and PDGFRA^{Lo} cells separately (Supplementary Fig. 1g).

- What is the additional information that can be derived from the PDGFRA-CreERT2/Rosa26-mTmG experiment? Please elaborate on this.

Response: The only point we intended to make with the mTmG experiment is that cytoplasmic fluorescence allowed us to examine the morphology of PDGFRA⁺ cells (this is not possible in *Pdgfra*^{H2B-eGFP} mice, where only cell nuclei are fluorescent), as shown in Fig. 1e.

- Fig 1g and Suppl Fig 1g: In my view, the data of Suppl. 1g is more interesting to the story and should be flipped with the data of Fig 1g. What happens if Wnt is left out of the medium in the PDGFRA low co-cultures? The wording “from different gastric segments” is a bit unclear, why not write “from antrum and corpus”?

Response: We appreciate this suggestion. The original Fig. 1g and Supplementary Fig. 1g are now combined in revised Fig. 2b.

Reviewer 2

Using high-resolution imaging and sequencing, Manieri and colleagues characterized subpopulations and niche properties of mouse corpus and antral PDGFRA⁺ cells. They report that PDGFRA^{Lo} CD55⁺ cells enable corpus and antral organoid growth in contrast to CD55⁻ cells. The study shows a comprehensive characterization of CD55⁺PDGFRA^{low} cells. While this study is mostly descriptive, a little more depth would be required to strengthen the claims and better understand the function of these cells within the stem cell niche. Major points are described below.

Response: We thank the Reviewer for a careful reading and for constructive suggestions to improve the study.

1. *Grem1* expression is quite low in PDGFRA^{low} cells in Antrum and Corpus, and other cells have been described to have prostemness niche function, for example lymphatics. Are CD55⁺ cells the only population to have niche function in their model, including other cells close to stem cells such as PDGFRA⁻ and CD31⁺ populations?

Response: This is an important question. We extended the study by assessing the activities of antral and corpus CD31⁺ (blood and lymphatic endothelial) cells and PDGFRA⁻ mesenchyme in gastric gland co-cultures. Unlike PDGFRA^{Lo} cells, neither population induced spheroid formation (Suppl. Fig. 3b). Importantly, we regard *Grem1* and *Rspo3* (which express at lower levels in stomach than in intestinal mesenchyme) as useful molecular markers, not necessarily as crucial signals. Indeed, new data in the revised manuscript reveal subtle effects upon ablation of gastric *Grem1*⁺ cells, in contrast to the drastic effects upon ablation of intestinal *Grem1*⁺ cells (Fig. 7b).

2. Addition of stromal cells to intestinal organoids has been shown to induce spheroids. Analysis and quantifications *in vitro* should discriminate between organoids and spheroids.

Response: Except for the effect of rFGFs on “budding” (of unclear significance), all structures we observed in gastric gland co-cultures in this study were spheroidal in morphology, similar to the morphology of intestinal crypt co-cultures and distinct from that of intestinal crypts cultured with recombinant factors. The revised manuscript therefore eschews the misleading use of the term “organoid,” except to describe the *in vitro* assay in the generic sense of current usage.

3. On which factor depend the effect of CD55⁺ cells on organoids/spheroids? Is this effect dependent on *Grem1*?

Response: This is an important question, which we addressed in two ways. First, we introduced the Porcn inhibitor Wnt-C59 (to impair Wnt secretion) or a cocktail of BMP ligands (to negate the effects of *Grem1* and other BMPi) in co-cultures of gastric glands with CD55⁺ PDGFRA^{Lo} cells (new Fig. 6f). Second, we treated *Grem1*^{DTR} mice with Diphtheria toxin to ablate the small *Grem1*⁺ subpopulation of CD55⁺ PDGFRA^{Lo} cells *in vivo* (Fig. 7a-b). Although BMPs inhibited spheroid formation *in vitro*, selective ablation of *Grem1*⁺ cells had subtle effects *in vivo*, indicating that CD55⁺ cells other than the minority *Grem1*⁺ fraction provides a signal that BMPs can override *in vitro*. We discuss the implications of these findings (p. 13).

4. To clarify the function of this population, it would be useful to provide additional evidence that these cells are essential and required to maintain stem cells *in vivo*. This could be done by performing depletion studies for example with *LepR* reporter mice, as *LepR* seem to be expressed in these cells.

Response: The Reviewer raises a fair point and suggests a useful path. However, our scRNA analysis showed limited overlap between *LepR* and *Cd55* transcripts, with *LepR* expressed in many *Cd55*⁺ cells and absent from CorpLo1, the corpus *Pdgfra*^{Lo} population with highest *Cd55* expression (see below). Therefore, even if *LepR*⁺ cell depletion has consequences, we currently lack a framework to interpret the findings. In contrast, *Grem1*⁺ cells represent a distinct subset

of the *Cd55*⁺ population and *Grem1* is not significantly present in other cell types. Therefore, to address the broader question from Reviewer 2 and others, we treated *Grem1*^{DTR} mice with Diphtheria toxin to deplete *Grem1*-expressing cells *in vivo* (p. 12 and new Fig. 7a-b).

5. A population of PDGFR^{low}CD55⁺ that express *Grem1* has already been reported (Jasso et al., 2022), although in the colon. How similar are these populations in terms of gene signature?

Response: On the Reviewer's recommendation, we compared PDGFR^{Lo} CD55⁺ cells with the colon population reported by Jasso et al., 2022. Using stringent criteria (\log_2 fold difference > 0.5 and pct. expressed cells ratio > 2), we derived gene signatures for corpus and antral CD55⁺ cells distinct from CD55⁻ cells. As we now mention in the Discussion (p. 13), these signatures showed good overlap with the 'Fibroblast 3a' population from the study by Jasso et al. The summary shown below for the Reviewer's interest is not included in the revised manuscript.

6. CD55 expression has been described on several other cell types. The claim that CD55 alone is sufficient to identify niche stromal cells should therefore be further investigated. Detailed FACS analysis for CD55 in different subsets of CD45⁺ (including myeloid and lymphoid gates) and CD45⁻ cells (including PDGFRA⁺ and PDGFRA⁻ cells) and epithelial cells from the lamina propria could help strengthen this claim.

Response: This is also an important question. We did not identify CD55 expression in any native mesenchymal cell population other than a fraction of PDGFRA^{Lo} cells (Fig. 5c); however, CD55 is expressed in some CD45⁺ (PTPRC⁺) non-native cells. We used flow cytometry to identify the expressing fraction among the ~26% of CD45⁺ cells in the corpus and antrum. Most of these are CD11b⁺ myeloid cells (corpus 40%, antrum 49%), followed by CD3⁺ T lymphocytes (corpus 20%, antrum 12%) and CD19⁺ B cells (corpus 5%, antrum <2%). Only B lymphocytes express CD55 at levels similar to sub-glandular PDGFRA^{Lo} cells (p. 10 and new Supplementary Fig. 7c).

7. CD55 is a surface marker, which can be upregulated and downregulated on niche cells. What regulate CD55 expression on these cells? Any evidence that expression of CD55 is faithful to this population at different ages and conditions (notably inflammation)?

Response: Our study reports the discovery and value of CD55 as a surface marker to isolate mesenchymal cells with niche activity *in vitro*. We do not suggest that CD55 function itself is relevant to niche function and CD55 regulation is therefore well outside the scope of the present manuscript. Nevertheless, we addressed the Reviewer's question about CD55 expression with respect to mouse age and found similar levels by flow cytometry in PDGFRA^{Lo} cells from two 3-week-old and one 1 year-old mouse. Because the sample size is small, we do not include these data in the revised manuscript but show them below for the Reviewer's interest.

8. What is the lineage relationship with the other subsets of PDGFRA⁺ and other progenitor cells that have been described in the lamina propria? Pseudotime analysis could help answer this question.

Response: In the absence of experimental evidence that any adult mesenchymal cell population gives rise to any other, pseudotime trajectories have a high risk to be misleading. Pseudotime is useful in ordering cells from prior biological knowledge, with limited utility in *de novo* discovery of trajectories and especially in determining directionality without prior knowledge (e.g., Weinreb et al., 2018). Nevertheless, we imputed RNA velocity (Bergen et al, 2019) by computing the latent time (average position of each cell along inferred transcriptional dynamics of all genes) on corpus and antral PDGFRA⁺ cells. *Pdgfra*^{Lo} and AntInt contained a higher proportion of cells with low latent time (i.e., early in the inferred developmental trajectory) than *Pdgfra*^{Hi} cells. As the biological significance of these findings is unclear (and may stir needless controversy), we do not include them in the revised manuscript but show them here for the Reviewer's interest.

9. More information on the crosstalk of this population with surrounding cells would help increase our knowledge of the niche, notably with immune and vascular cells.

Response: On the Reviewer's recommendation, we used a curated database of mouse ligand-receptor pairs in Cellchat (Jin et al., 2021), averaged the expression of ligands and receptors for each pair (accounting for stoichiometry of signaling complexes) in each cell type, and identified putative interactions between pairs of cell types. The numbers and summed strengths (high ligand or receptor RNA expression) provide hypotheses for potential cell-cell interactions. Antral and corpus *Pdgfra*^{Hi} and *Pdgfra*^{Lo} cells, myocytes, and glial cells showed signaling connectivity, possibly forming a 'module' of coordinated cell types. Results are shown here for the Reviewer's interest but are not included in the manuscript because the analysis is "hypothesis-generating" and inherently speculative. Investigation of one or more hypotheses would merit its own study, well beyond the scope of the present manuscript.

10. Is this population found in the human gastric mesenchymal niche, at steady-state and in chronic pathologies such as gastric cancer?

Response: This is an interesting question. To determine whether cells we describe are present in human gastric mesenchyme and have a role in disease, we visited scRNA-seq data (Kang et al, 2022) on gastric cancers and matched non-malignant gastric tissue from 24 treatment-naïve subjects. We used the gene signatures derived above (see response to Comment #5) for antral and corpus CD55⁺ cells and scored cells in the gastric cancer dataset according to these signatures. Fibroblasts in gastric cancers had higher signature scores compared to other cell types in the tumor microenvironment and the resting fibroblast population that Kang et al. call 'F01' fibroblast expresses these signatures more highly than other subtypes. The revised manuscript discusses these similarities (p. 13).

Reviewer 3

Manieri et al., 2023 characterized the mesenchymal niche of the murine gastric epithelium which has not been thoroughly studied so far. The authors divided the corpus and antrum regions and compared the mesenchymal profiling of the corpus, antrum, and small intestine to comprehensively characterize each mesenchymal niche. Similar to their previous study (McCarthy et al., 2020), imaging, bulk RNA sequencing of PDGFRA⁺ cells, single-cell RNA sequencing, in situ hybridization, and organoid experiments were used. They discovered that the PDGFRA⁺ cells, particularly the PDGFRA [low] cells, are sufficient to support epithelial stem

cell growth in both corpus and antrum, while the PDGFRA [high] SEMF cells express potential niche signaling elements. They finally found out that Cd55, a membrane protein, is expressed in the PDGFRA[low]-specific manner, and postulated that it could be a useful molecular marker for isolating PDGFRA[low] cells in both the corpus and antrum.

Even though the data, including the scRNA-seq, is informative and has great potential to be a useful resource in the field, the current manuscript has a significant number of concerns and critical points that need to be addressed.

Response: We appreciate the Reviewer's measured critique and specific recommendations.

1. The majority of references regarding stem cell populations in the stomach are absent. Especially, this manuscript only cites Karam, S.M.'s 1993 papers for the introduction of the corpus stem cells. Consequently, the current description of stem cells is wrong and the interpretation of their data is too simplistic. Many parts must be revised or rewritten in light of the following papers. As the manuscript defines the niches for stem cells and differentiated cells, the precise introduction of the most up-to-date information regarding stomach stem cells is required. In the corpus epithelium, two distinct stem cell populations have been identified: fast-cycling isthmus stem cells and the most-quiescent chief cells (PMID: 31422913, PMID: 31589873). The isthmal stem cells control the pit, isthmus, and upper neck regions, while the chief cells control the lower neck and chief cell regions in homeostasis (PMID: 31422913, PMID: 31589873). In homeostasis, the chief cells express Wnt pathway genes, such as Troy (PMID: 24120136) and Lgr5 (PMID: 28581476); upon injury, the chief cells become activated to replenish the cell loss (PMID: 20854822, PMID: 24120136, PMID: 28581476, PMID: 32768422, PMID: 29467218, PMID: 34497145, PMID: 35523142).

Given this information, the following parts in the introduction should be particularly revised: In lines 41-43, 'whereas the gastric corpus epithelium turns over slowly, ~in the isthmus of the each glandular unit' is the wrong description.

Line 45 'Antral stem cells express Lgr5 and concentrate near the gland base' misses the citation (Barker et al., 2010; PMID: 32025032). A recent paper about antral stem cells is also missing (PMID: 32025032).

I believe the authors could analyze their data and extract more intriguing observations based on the background information from the missing references. The current version does not investigate the corpus mesenchymal niche in depth due to the absence of these references.

Response: We are grateful for this critique, which is most helpful. We have revised the Introduction extensively to incorporate the suggested content and previously missing references and to provide fuller context for the findings from our study.

2. The title 'Defining the structure, signals, and cellular elements of the gastric mesenchymal niche' could mislead the readers. As the manuscript is mainly focusing on the PDGFRA+ niche, the addition of the PDGFRA+ niche in the title should be better.

Response: We appreciate the suggestion and have changed the title to better reflect the theme and conclusions of the manuscript.

3. In Fig 1g, they argued that 'SI and gastric stem cells have distinct cellular support requirements', but their experiments are not enough to support this argument. 1) They did not assess if antral or corpus glands can form the organoids when supported by SI PDGFRA [low] cells. 2) There is a possibility that other niche stromal cells (or a mixture of other niche stromal cells with PDGFRA [low] cells from heterologous origin) can support the heterologous epithelial stem cells but it was not addressed as well. Unless these experiments are performed, it is hard to tell if the cellular support requirements are distinct.

Response: The conclusion about 'distinct cellular support requirements' is indeed superficial. We respond in two ways. First, the revised manuscript includes the requested data showing that PDGFRA^{Lo} cells from SI support spheroid formation from corpus and antral glands (Fig. 2b) and that CD31⁺ endothelial or PDGFRA⁻ cells fail to support spheroid formation (Supplementary Fig. 3b). Second, we have modified the wording for a more conservative conclusion (p. 4).

4. They used the same media cocktail (ENRW) for SI organoids and stomach organoids. However, it is well known that the stomach organoids need two more components, Gastrin and Fgf10 for their maintenance (PMID: 33223522). This is also shown in the reference they cited (Barker et al., 2010). They should change the media condition to the right one.

Response: We used standard media in all experiments: ENR (EGF, Noggin, Rspo1 conditioned medium – no Wnt) for SI organoids and ENRWG (EGF, Noggin, Rspo1 conditioned media, Afamin/Wnt3a conditioned medium, and Gastrin) for gastric spheroids. In the original Barker et al study (*Cell Stem Cell* 2010), hFGF10 served only in gastric organoid budding, which is not relevant to our main questions, so in the interest of simplicity we deliberately omitted hFGF10 (we do, however, quantify spheroid budding with hFGF10 and mFGF7 in Fig. 4e). Neglecting to mention W and G in the gastric spheroid recipe was an error in our first version and is corrected in the second (p. 16). As specified in the manuscript, all experiments to define mesenchymal cell activity occurred in basal medium lacking exogenous EGF, Rspo, Noggin, Wnt, Gastrin or FGF).

5. They did not properly describe what the 'organoid number' and 'organoid area' mean (Fig 2, Extended Data Fig. 2) and how the quantification was performed. How did they match the starting number of cells? Did they isolate the single cells and seed the same number of cells in each dome? Are organoid area and organoid size the same? If the organoid area means the total area taken by all the organoids, the organoid area can be affected by the number of organoids. Is this the best measurement tool to show the difference in growth and forming efficiency of the organoids?

Response: As Reviewer 1 recommended, and because detailed experiments with different organoid (spheroid) media distracted from the narrative flow without contributing to the study, we have removed those data from the revised manuscript. Accordingly, we no longer present 'organoid area.' All comparisons in the study are based on plating equal numbers of gastric glands in media with growth factors or mesenchymal cells.

6. In Fig. 3, scRNA seq was performed on all stromal cells in the corpus and antrum; however, only pdgfra⁺ populations were analyzed further because they displayed a lower degree of similarity in pdgfr⁺ populations between the corpus and antrum (Fig. 3b). Indeed, the UMAP

clearly separates the *pdgfra*⁺ cells in the corpus and antrum (Fig. 3c). Nonetheless, this type of distinct separation may occur if the batch effect of the experiment is not corrected. If cells are processed separately on distinct chips, the batch effect can occur. The authors should explain in the methods section how they corrected the batch effect and how they came to the conclusion that the separation of *pdgfra*⁺ populations between the corpus and antrum is based on their actual characteristics.

Response: This is an important technical point. To address possible batch effects, we compared corpus and antral cell types for consistency, both by marker expression and by MetaNeighbour (Crow et al., 2018) analysis (Fig. 3b). MetaNeighbour constructs a weighted-neighbor-voting classifier for each pair of cell clusters and uses the efficacy of the classifier, quantified by the AUROC (area under the receiver operating curve) score as a metric of cell type similarity. Good correlations between the same cell types in corpus and antrum provide evidence for the consistency of cell types isolated from the two tissues and argue strongly against a batch effect.

To address any concerns about separation of *Pdgfra*⁺ populations, we conducted an additional analysis using AUGUR (Squair et al., 2021), which uses a regularized random forest classifier to quantify transcriptional differences between cell types under different conditions (e.g., corpus vs. antrum). High AUC scores indicate larger transcriptional difference between a given cell type in corpus vs. antrum. *Pdgfra*^{Hi} had a significantly higher AUC score, i.e., larger transcriptional differences, compared to other cells, affirming that the differences we report are biological rather than technical effects (batch effects are unlikely to affect different populations within the same cell isolates to such different degrees). AUGUR analysis is reported in Supplementary Fig. 4c.

7. The result section of Fig. 3c states that the *Pdgfra*⁺ cells contain 6 clusters, but the UMAP in Fig. 3c displays only 4 colors (purple, red, blue, and green). What are these colors based on? Authors should include a suitable annotation and proper explanation of this mismatch.

Response: Our initial use of colors and cluster designations was indeed confusing and we thank the Reviewer for pointing that out. In the revised manuscript we adopt consistent nomenclature and cluster designations as the narrative unfolds.

8. Even though the authors discovered multiple DEGs in each cluster using scRNA-seq, they validated just a handful of genes. Specifically, there are no validation data in the corpus region, and the researchers stated that RNA Scope failed ('In situ hybridization was unreliable in the corpus' in line 308). As they still have the option to use IHC, they should exert more effort to provide readers with beneficial information regarding the spatial expression pattern of the potential marker genes.

Response: This is a fair criticism, similar to the only major concern voiced by Reviewer 1. On their recommendation, we overcame previous technical limitations with RNA in situ hybridization (ISH) in corpus tissue (although immunohistochemistry with multiple CD55 antibodies remained uninterpretable owing to high background). ISH probes for CD55-coexpressed markers *Grem1* and *Rspo3* localize corpus CD55⁺ PDGFRA^{Lo} mesenchymal cells to the space between gland bases and the muscularis mucosae (new Fig. 5d), similar to our findings in the antrum (Fig. 6a).

9. The authors divided the antral SEMF (*Pdgfra* [high]) into two populations: Fgf7+ Nrg1+ SEMF

in the upper glands adjacent to pit cells and Bmp3+ Ctgf+ SEMF in the lower glands adjacent to proliferative cells. Given that the Bmp3+ Ctgf+ expressing-SEMF subpopulation is in close proximity to the proliferative cells, the authors sought to determine whether SEMF can promote antral gland growth (proliferation) in vitro in Fig. 4d. The PDGFRA[high] cells (GFP[High] cells from Pdgfra[H2BeGFP] antrum) were cocultured with the antral glands due to the inability to identify an appropriate surface marker isolating Bmp3+ Ctgf+ SEMF exclusively. Due to two main factors, they cannot reach the conclusion stated on pages 255-256 using this experiment.

- 1) PDGFRA[high] cells must be Fgf7+ Nrg1+ SEMF and Bmp3+ Ctgf+ SEMF positive. As Fgf7+ Nrg1+ SEMFs are found in close proximity to differentiated pit cells, they may function to promote epithelial cell differentiation. Therefore, it is not possible to rule out the possibility that Fgf7+ Nrg1+ SEMF reduces the effect of Bmp3+ Ctgf+ SEMF.

- 2) The author demonstrated that PDGFRA[high] cells alone are insufficient to sustain stem cells. If there is no niche factor for stemness, then organoids cannot form in the first place; consequently, there is no organoid growth (proliferation). Therefore, the authors must redesign the experiment to include minimal niche factors (Wnt agonists) and investigate the difference in proliferation according to the presence of Bmp+ Ctgf+ SEMF.

Response: The conclusion to which the Reviewer refers is indeed stronger than our experiment with mixed PDGFRA^{Hi} cells allows; antral SEMF subpopulations we identified could play distinct (and possibly opposing) roles, but as we noted and the Reviewer acknowledges, no surface marker currently allows their separation. We have deleted the previously flawed statement. More importantly, we have also now included specific growth factors or cocktails in co-culture experiments with unfractionated antral and corpus SEMFs. In the presence of SEMFs, addition of Rspo1 and Wnt3a increased spheroid formation to the levels seen with complete media (Supplementary Fig. 5d), suggesting that SEMFs lack at least the signals represented by these additives.

10. What is the connection between the subpopulations in Figure 3c (CorpLo, CorpLo1, AntInt, AntLo) and Figure 5a (CorpLo1, CorpLo2, CorpLo3, AntInt, AntLo)? As these subclusters were extracted using the same scRNA seq data, there may be a correlation between them. Otherwise, they must annotate each cluster with a unique name so that readers can comprehend the data correctly.

Response: We apologize for the confusion resulting from our inconsistent use of colors and cluster designations in early and later figures in the original manuscript. We have now modified the names and color designations of each population to ensure consistency and flow.

Minor points

1. Proper annotation including cell types in the corpus and antrum glands in Fig 1a is needed.

Response: This was an error in our final preparation of the figure and is now rectified (Fig. 1a).

2. Laminin staining cannot directly show the gland pit. Pit cell marker staining using Muc5ac antibody or UEAI will be the proper marker for pit cells.

Response: The reviewer is correct. Laminin staining outlines the foveolar opening but does not reveal foveolar depth. The revised manuscript includes stains with UEA-I and GS-II lectins to highlight pits as well as neck (corpus) and mucous basal cell (antrum) zones in the glands (Supplementary Fig. 1b).

3. Errors in the Figure citation:

- in line 81, Fig. 1a -> Fig. 1d
- in line 111, Fig. 2a -> Extended Data Fig. 2a

Response: Thank you for noticing these errors, which we have corrected.

4. In lines 214-216, the *Wnt4* transcript level is not shown in the corresponding figures.

Response: We apologize for the omission. *Wnt4* transcript levels are now shown in (Supplementary Fig. 4d).

5. The remaining GFP in Fig. 5c cannot correctly reflect the original GFP expression. Authors could use GFP probes to show GFP expression levels.

Response: Target retrieval steps in the RNA in situ hybridization (ISH) protocol extinguish native GFP signal. We therefore used GFP antibody to identify GFP^{Hi} SEMFs and weakly stained cells present in the same locations as native *PdgraLo* cells. This is now clarified in the methods (p. 20).

REVIEWER COMMENTS

Reviewer #1 (Remarks to the Author):

In the revised manuscript, Manieri et al. responded to the major point raised by this reviewer, the missing localization of Grem1 and Ctgf in the corpus region of the stomach. The authors now added new data on the localization of these cells in the corpus, and also elaborate on the finding by proposing different models that take into account the distance between the cells and the active isthmus stem cells and the proximity to “facultative” stem cells at the gland bottom. Furthermore, in vitro data of co-culture assays were added, in which the authors examined the role of Wnt or BMP inhibition within the CD55+ cell fraction (for both the antrum and the corpus), highlighting the functional role of these signaling pathways with these cells. In addition, all minor points raised were addressed to satisfaction of this reviewer.

Reviewer #2 (Remarks to the Author):

The authors have made significant improvements to the manuscript to support their claims. However, a number of questions remain unanswered/unclear.

- Authors should show in situ hybridization for CD55 in addition to ISH of signature genes (Fig 5)

- The authors made clear that CD55 is only a marker in this study, and did not select CD55High cells for their study (Supp Fig 7a and 7b). Nevertheless, high/low expression levels of CD55 are mentioned several times in the results/discussion. This should be clarified, or authors should provide data showing that GFP^{low}CD55^{High} cells have a stronger capacity for spheroid formation than GFP^{low}CD55^{Low} cells.

- This is also true for the conclusion/legend of Supp Fig 7c, which can be misleading.

- Point 6 has been partially answered, however there is no information about CD55 expression in PDGFR α - cells and epithelial cells.

- Point 7: For CD55 to be of value as a surface marker to isolate mesenchymal cells with niche activity, it should be stable. It has been reported that CD55 expression/shedding is modulated by cytokines (Qiao et al., Oncotarget 2018; Blok et al., 2003; Nasu et al., 1998). Any evidence on gastric mesenchymal cells?

- Line 309: It is noteworthy that CD34 also marks SI mesenchymal cells with niche functions (Stzepourginski et al., 2017)

Reviewer #3 (Remarks to the Author):

All previous comments were addressed in full. With the addition of a proper introduction and a comprehensive analysis of scRNA seq, the revised manuscript has considerably improved! This manuscript contains vital information regarding the gastric stromal niche. Only two parts from newly added data need to be revised. Once the authors have addressed the following

comments, I am in favor of publishing the revised manuscript.

In Supplemental Figure 1a, the authors have added a summary table of the markers they used. The cell type and structure description for Tnfrsf19 (Troy) is incorrect. Even though a small fraction of parietal cells -at the base region and a rare subpopulation of isthmal cells are known to express Troy, it is generally accepted that Troy expression is most prominent in the chief cells of the corpus glands (PMID: 24120136).

2.The quality of the new H&E data in Supplemental Figure 2a is poor. It is nearly impossible to distinguish the glandular cell types.

Response to Reviewer Comments

Reviewer 1

In the revised manuscript, Manieri et al. responded to the major point raised by this reviewer, the missing localization of *Grem1* and *Ctgf* in the corpus region of the stomach. The authors now added new data on the localization of these cells in the corpus, and also elaborate on the finding by proposing different models that take into account the distance between the cells and the active isthmus stem cells and the proximity to “facultative” stem cells at the gland bottom. Furthermore, *in vitro* data of co culture assays were added, in which the authors examined the role of Wnt or BMP inhibition within the CD55+ cell fraction (for both the antrum and the corpus), highlighting the functional role of these signaling pathways with these cells. In addition, all minor points raised were addressed to satisfaction of this reviewer.

Response: We thank the Reviewer for the favorable evaluation.

Reviewer 2

The authors have made significant improvements to the manuscript to support their claims. However, a number of questions remain unanswered/unclear.

1. Authors should show *in situ* hybridization for CD55 in addition to ISH of signature genes (Fig 5).

Response: We value this suggestion. *In situ* hybridization with new *Cd55* probes (RNAscope) is now reported in Figs. 5d (corpus) and 6a (antrum). We interpret the results in conjunction with scRNA-seq data.

2. The authors made clear that CD55 is only a marker in this study, and did not select CD55^{High} cells for their study (Supp Fig 7a and 7b). Nevertheless, high/low expression levels of CD55 are mentioned several times in the results/discussion. This should be clarified, or authors should provide data showing that GFP^{low}CD55^{High} cells have a stronger capacity for spheroid formation than GFP^{low}CD55^{Low} cells.

- This is also true for the conclusion/legend of Supp Fig 7c, which can be misleading.

Response: We are a bit puzzled by these comments because a search of the manuscript (using the terms “CD55” and “high” or “hi”) turned up no instance of high/low CD55 expression *per se*. Perhaps the Reviewer is referring to our laboratory’s previous identification (PMID: 37028407) of CD55 as an intestinal mesenchyme marker, where *Cd55* RNA is expressed along a gradient and, to be conservative, we referred to CD55^{Hi} and CD55^{Lo} cells? In the stomach, mesenchymal cells are reliably CD55+ or CD55-, as the manuscript reflects. Nevertheless, we have deleted all potentially confusing associations of CD55 with the word “high” in lines 427 (previously line 420) and in the legends to Fig. 5c and Supplementary Fig. 7e (previously Supplementary Fig. 7c).

3. [Previous] point 6 has been partially answered, however there is no information about CD55 expression in PDGFR α - cells and epithelial cells.

Response: The revised manuscript provides additional data on CD55 expression in epithelial and in Pdgfra- mesenchymal cells (Fig.7d-g, Results line 312).

4. [Previous] point 7: For CD55 to be of value as a surface marker to isolate mesenchymal cells with niche activity, it should be stable. It has been reported that CD55 expression/shedding is modulated by cytokines (Qiao et al., Oncotarget 2018; Blok et al., 2003; Nasu et al., 1998). Any evidence on gastric mesenchymal cells?

Response: Our data reveal sufficient stability of CD55 expression on mesenchymal cells to allow routine isolation of the CD55+ cell fraction by FACS (Supplementary Fig. 7c). In contrast, stomach epithelial cells express *Cd55* RNA, detected by in situ hybridization (Figs. 5d and 6a) and scRNA-seq (Suppl. Fig. 7a), but do not express surface CD55 protein, as determined by flow cytometry (Suppl. Fig. 7b). Whether the latter discordance reflects inefficient RNA translation or CD55 modulation by cytokines (as the Reviewer asks and the revised Discussion addresses, lines 437-439) is not material to the present study and best reserved for future investigation focused on CD55 biology, distinct from its use as a molecular marker in our study.

5. Line 309: It is noteworthy that CD34 also marks SI mesenchymal cells with niche functions (Stzepourginski et al., 2017)

Response: Yes, we pointed this out in previous versions of the manuscript (lines 306-307) and cited appropriate references (#33 and #54). However, CD34 marks all intestinal PDGFRA^{Lo} cells and also endothelial cells, so it is less useful than CD55 in isolating mesenchymal cells that show activity in organoid assays.

Reviewer 3

All previous comments were addressed in full. With the addition of a proper introduction and a comprehensive analysis of scRNA seq, the revised manuscript has considerably improved! This manuscript contains vital information regarding the gastric stromal niche. Only two parts from newly added data need to be revised. Once the authors have addressed the following comments, I am in favor of publishing the revised manuscript.

1. In Supplemental Figure 1a, the authors have added a summary table of the markers they used. The cell type and structure description for *Tnfrsf19* (Troy) is incorrect. Even though a small fraction of parietal cells -at the base region and a rare subpopulation of isthmal cells are known to express Troy, it is generally accepted that Troy expression is most prominent in the chief cells of the corpus glands (PMID: 24120136).

Response: We regret this error, which is now corrected (Suppl. Fig. 1a).

2. The quality of the new H&E data in Supplemental Figure 2a is poor. It is nearly impossible to distinguish the glandular cell types.

Response: We apologize for the previously poor quality of H&E stains. These are now replaced with images of higher quality, which allow visualization of glandular cell types (Suppl. Fig. 2).

REVIEWERS' COMMENTS

Reviewer #2 (Remarks to the Author):

The authors made substantial revisions and answered to the major concerns. I support publication of the study, which should nevertheless still address and clarify the points below.

- Concerning point 2: If the authors do not have a functional meaning for differences in CD55 expression levels (as they report in Supp Fig. 7e), the percentage of CD11b, CD3 and B cells expressing CD55 should be clearly stated and reported in the result section (line 314, p.10). In addition, in these FACS for immune cells, the gate for viable cells (as shown in Supp Fig.10) should be adjusted to eliminate only DAPI+ cells (as in Supp Fig.9).

- The conclusion p13-14 remains quite puzzling. As the data show CD55 expression on CD45+ cells and epithelial cells (which can be difficult to discriminate by FACS), CD55 cannot be used as a single marker to isolate mesenchymal niche cells. This should be clarified.

Reviewer #3 (Remarks to the Author):

No more comments!

Response to Reviewer Comments

Reviewer #2

The authors made substantial revisions and answered to the major concerns. I support publication of the study, which should nevertheless still address and clarify the points below.

Response: We appreciate the support for publication.

Concerning point 2: If the authors do not have a functional meaning for differences in CD55 expression levels (as they report in Supp Fig. 7e), the percentage of CD11b, CD3 and B cells expressing CD55 should be clearly stated and reported in the result section (line 314, p.10).

Response: These results are now reported in the Results section (lines 312-313) as requested.

In addition, in these FACS for immune cells, the gate for viable cells (as shown in Supp Fig.10) should be adjusted to eliminate only DAPI+ cells (as in Supp Fig.9).

Response: We have made the requested adjustment and amended Supp Fig. 10 as requested.

The conclusion p13-14 remains quite puzzling. As the data show CD55 expression on CD45+ cells and epithelial cells (which can be difficult to discriminate by FACS), CD55 cannot be used as a single marker to isolate mesenchymal niche cells. This should be clarified.

Response: The requested clarification is included in the Discussion (lines 431-434). Please note that **(1)** epithelial cells express *Cd55* RNA but flow cytometry does not detect CD55 on epithelial cell surfaces (Supplementary Fig. 7b), and **(2)** epithelial cells are easily and routinely discriminated from mesenchymal cells by EPCAM flow cytometry. Therefore, CD55 is a reliable marker to isolate the mesenchymal cell fraction with demonstrable niche activity.

Reviewer #3

No more comments!

Response: We appreciate the support for publication.